# Riemannian Neural SDE: Learning Stochastic Representations on Manifolds

**Sung Woo Park**[1], **Hyomin Kim**[2], **Kyungjae Lee**[3], **Junseok Kwon**[4]

[1,2,4]School of Computer Science and Engineering, Chung-Ang University, Korea
[1,3,4]Artificial Intelligence Graduate School, Chung-Ang University, Korea
[1]LG AI Research
[1]pswkiki@gmail.com, [2]icecream126@cau.ac.kr
[3]kyungjae.lee@ai.cau.ac.kr, [4]jskwon@cau.ac.kr

## Abstract

In recent years, the neural stochastic differential equation (NSDE) has gained attention for modeling stochastic representations with great success in various types of applications. However, it typically loses expressivity when the data representation is manifold-valued. To address this issue, we suggest a principled method for expressing the stochastic representation with the Riemannian neural SDE (RNSDE), which extends the conventional Euclidean NSDE. Empirical results for various tasks demonstrate that the proposed method significantly outperforms baseline methods.

## 1 Introduction

Recently, there has been a great success in modeling stochastic dynamical systems for complex data representations with spatially high stochasticity. Particularly, recent studies have utilized the *stochastic differential equation* (SDE) as a fundamental probabilistic model to express the transition of stochastic states. The pioneering work, *Latent SDE* (14), introduced the deep-learning framework to utilize SDE for modeling spatio-temporal representations. Following their work, parameterized reverse-SDE (1) was adopted to model score-based generative models (27). Recently, a controlled SDE combined with a stochastic optimal control theoretical framework was introduced to model time series (21). These studies provided a new

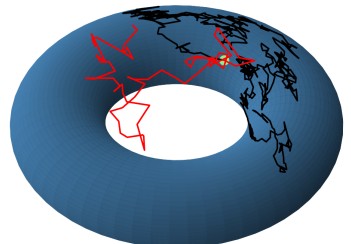

**Figure 1:** Brownian motions on torus $\mathbb{T}^2$ (**black**) and $\mathbb{R}^3$ (**red**).

method for describing stochastic representations and illustrated the superiority of neural SDE models in various real-world applications. Although remarkable progress has been recorded in recent studies, the primary interest in using SDE has focused on Euclidean geometry (*i.e.*, $\mathbb{R}^d$). Thus, conventional approaches inevitably lose their expressivity when data representation is defined in generic geometry, such as Riemannian manifolds (*i.e.*, $\mathcal{M}$). In this paper, we solve the aforementioned problem by introducing the *Riemannian neural SDE* (RNSDE), which can model the stochastic representation on manifolds. The proposed RNSDE is a natural extension of conventional Euclidean SDEs that defines intrinsic stochastic transitions while fully considering the geometric structure. Specifically, we follow the *Eells-Elworthy-Malliavin* interpretation of the diffusion Riemannian SDE, where the stochastic representation is expressed on the frame bundle.

**Example.** To illustrate the problem mentioned above, we represent the stochastic trajectories of Riemannian (**black**) and Euclidean (**red**) Brownian motions on 2-Torus in Fig.1. As shown in the figure, the Euclidean SDE is unaware of the underlying geometry (*i.e.*, torus) and its trajectory easily steps out of the curved surface. Accordingly, the use of Euclidean SDE is problematic and lacks geometry-awareness. In contrast, the proposed Riemannian SDE respects its underlying geometry and yields a trajectory that clearly lies on the manifold.

36th Conference on Neural Information Processing Systems (NeurIPS 2022).

**Contribution.** The main contribution of this work is to suggest a general framework for modeling the stochastic representations on Riemannian manifolds. Specifically, we define the diffusion process on manifolds based on the Eells-Elworthy-Malliavin interpretation. Given well-defined Riemannian SDEs, we introduce a novel Markov diffusive Kantorovich dual formulation and suggest a gradient flow-based algorithm to train the neural network.

**Notation.** Einstein's summation conventions are used throughout the paper. The Euclidean and Riemannian norms are denoted as $\|u\|$, $\|v\|_g$ with vectors $u \in \mathbb{R}^n$, $v \in T_{(\cdot)}\mathcal{M}$, respectively. $\partial_{x_i} := \frac{\partial}{\partial x_i}$ and $\partial_t := \frac{\partial}{\partial t}$ denote spatial and temporal partial derivatives, respectively. We consider the probability space $(\mathcal{M}, \Sigma, \mathbb{P})$ where the filtration $\{\mathcal{F}_t\}_{0 \leq t \leq T}$ is augmented by the Brownian motion $\{B_t\}_{t \in [0,T]}$ in the time interval $[0,T]$.

**Riemannian geometry.** In this paper, we consider complete and oriented $n$-dimensional Riemannian manifolds $(\mathcal{M}, g)$ equipped with Riemannian metric $g$. In particular, we focus on embedded compact sub-manifolds $\mathcal{M}$ (*e.g.*, sphere and torus) of the ambient Euclidean space, $\mathcal{M} \subseteq \mathbb{R}^D$, for $n \leq D \leq 2n$[1]. The metric tensor is expressed in the matrix form $G := [g_{ij}]_{1 \leq i,j \leq n}$. Similarly, we define the Riemannian co-metric as the inverse of matrix $G$ as $G^{-1} := [g^{ij}]$. The Christoffel symbol with respect to metric $g$ is denoted by the tensor form $[\Gamma^i_{jk}]_{1 \leq i,j,k \leq n}$. For the tangent space $T_{X_t}\mathcal{M}$ along the sample path, we denote $\{\partial^t_i\}_{1 \leq i \leq n}$ as the moving frame. The Riemannian volume measure is denoted as $d\mathbb{V} = \sqrt{|\det G(x)|}dx$. The orthogonal frame bundle and its corresponding local coordinate function are denoted as $O\mathcal{M}$ and $\psi : U \subset \mathbb{R}^n \to \mathcal{M}$, respectively.

## 2 Riemannian Neural Stochastic Differential Equation

We introduce a novel Riemannian SDE that defines the stochastic representations on Riemannian manifolds. Additionally, explicit local representations and a numerical scheme for implementation are presented. We begin by introducing the conventional neural SDE and extend the discussion.

**Euclidean Neural SDE.** The major interest of neural SDE is to train the parameterized stochastic process $X^\theta_t$ defined as a solution to the following Stratonovich SDE (20):

$$dX^\theta_t = h\left(t, X^\theta_t; \theta\right) dt + \sigma\left(X^\theta_t; \theta\right) \circ dB_t, \tag{1}$$

where $B_t = [B^1_t, \cdots, B^n_t]$ denotes the $n$-dimensional standard Brownian motion on $\mathbb{R}^n$, and Euclidean drift, and diffusion functions (*i.e.*, $h : [0,T] \times \mathbb{R}^d \times \Theta \to \mathbb{R}^d$ and $\sigma : \mathbb{R}^d \times \Theta \to \mathbb{R}^{d \times n}$, respectively) are parameterized by the neural network $\theta \in \Theta$. Euclidean SDE has obvious limitations in handling non-Euclidean datasets, because it produces outputs in ambient space. Therefore, we propose Riemannian neural SDE, which notably respects its underlying geometry.

**Riemannian Neural SDE.** Similar to Euclidean NSDE, our goal is to suggest a differential equation on manifolds. Specifically, we suggest the *Riemannian Neural Stochastic Differential Equation* (RNSDE) parameterized by two novel terms (*i.e.*, **(A)** and **(B)**), defined as follows:

$$dX^\theta_t = \underbrace{W\left(t, X^\theta_t; \theta\right) dt}_{\textbf{(A)}} + \underbrace{\beta(\theta)\pi^{-1}\left(X^\theta_t\right) \circ dB_t}_{\textbf{(B)}}. \tag{2}$$

**(A) Neural Potential Field.** In the first term, we refer to the parameterized vector field $W(t, X_t; \theta) := w^j(t, X_t; \theta)\partial^t_j : [0,T] \times \mathcal{M} \times \Theta \to T_{X_t}\mathcal{M}$ as the *neural potential field*, where the set of orthonormal tangent vectors $\{\partial^t_j\}_{1 \leq j \leq n} := \{\frac{\partial}{\partial x_j}|_{X_t}\}_{1 \leq j \leq n} : \mathcal{M} \to T_{X_t}\mathcal{M}$ is the moving frame on the trajectories of stochastic dynamics. This term defines the deterministic drift on manifolds in respect to the inferred network decision $W(t, X_t; \theta)$, based on the current state information

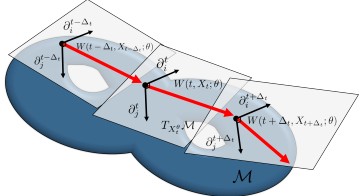

**Figure 2:** Neural potential fields.

$(t, X_t)$. The neural network $W$ produces tangent vectors (red arrows in Fig.2), which consecutively drive the stochastic trajectory to determine the next direction on $T_{X^\theta_t}\mathcal{M}$ (white planes in Fig.2).

While the Euclidean drift $h$ in (1) is limited to producing vectors in the ambient space $\mathbb{R}^D$, the neural potential field $W$ in (2) produces vectors in the tangent space, thus respecting the geometric surfaces.

---

[1]The Whitney's theorem (11) guarantees the existence of smooth embedding on $\mathbb{R}^{D=2n}$. Thus, we set the maximal dimension of $D$ to be bounded by $2n$.

**(B) Stochastic Development**. The second term, called *stochastic development* (8; 9), controls the diffusive/stochastic behavior of the proposed SDE on manifolds. The function $\pi : O\mathcal{M} \to \mathcal{M}$ is a canonical projection that maps the orthogonal frame bundle to model manifolds, where $U_t^\theta = \pi^{-1}(X_t^\theta) \in O\mathcal{M}$. This term projects the stochastic trajectory defined on the orthogonal frame bundle $U_t^\theta \in O\mathcal{M}$, which locally behaves like a **flat** Euclidean SDE, onto the **curved** model manifolds. This procedure is called *Eells-Elworthy-Malliavin* interpretation to present the SDE on the manifold. Intuitively, the stochastic development could be considered the procedure for rolling[2] the trajectory on a curved surface (*e.g.*, $\mathcal{M}$) over a flat plane (*e.g.*, Euclidean SDE in (1)). Meanwhile, the parameterized scalar-valued function $\beta : \Theta \to \mathbb{R}^+$ controls the diffusivity of the process $X_t^\theta$. It is noteworthy that the solution to (2) is the Brownian motion on $\mathcal{M}$, when $W \equiv 0$ and $\beta \equiv \frac{1}{2}$. This case is illustrated in Fig.1.

**Local Representations.** Although the proposed RNSDE in (2) is mathematically well-defined, it is not directly applicable to the numerical algorithm because of the abstract formulations of horizontal vector fields. Thus, for clarity, we provide explicit local representations expressed as Itô's SDE, as follows:

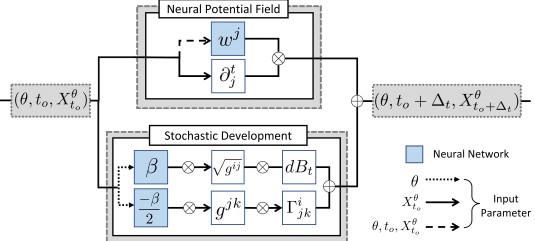

**Figure 3:** Local representations of the RNSDE.

**Proposition 1.** *(Local Representations) Let $X_t^\theta$ be a parameterized stochastic process which is the solution to the RNSDE in (2). In a local coordinate system, the proposed RNSDE can be rewritten as the Itô's representation:*

$$X_t^\theta = \beta(\theta) \int_0^T \sqrt{g^{ij}(X_s^\theta)} dB_s^j + \int_0^T \left[ \frac{-\beta(\theta)}{2} g^{jk}(X_s^\theta)\Gamma_{jk}^i(X_s^\theta) + w^j(X_s; \theta)\partial_j^s(X_s^\theta) \right] ds. \quad (3)$$

The detailed derivation is provided in Appendix C.3. The local-coordinate-based representation in (3) simultaneously requires geometric terms including metric tensor $g$, Christoffel symbol $\Gamma_{jk}^i$, and frame $\partial_j^s \in \mathcal{M}$, along the sample path $X_s^\theta \in \mathcal{M}$, that is calculated during sample propagation. The effect of stochastic development can be inferred from the newly appearing terms (*i.e.*, the first and second terms in (3)). It transforms the Euclidean diffusion induced by standard Brownian motion $B_t$ into Riemannian diffusion on manifolds by considering the curvature effect (*i.e.*, $g, \Gamma$).

**Euler-Maruyama Scheme.** To approximate the solution to the RNSDE in a local coordinate system, we apply the Euler-Maruyama scheme to simulate the SDE in (3). Let $t \in \mathbb{T} := \{t_o\}_{1 \le o \le T/\Delta_t}$ be the total simulation time and $\Delta_t$ be a priorly fixed time interval. This scheme provides sampled stochastic particles approximated as $\{X_{t_o + \Delta_t}\}_{1 \le o \le T/\Delta_t} \propto \mathcal{N}(\mathbf{m}_t \Delta_t, \sqrt{g^{ij}(X_s^\theta)\Delta_t} I_n)$, where the mean $\mathbf{m}_t \approx -\beta g^{jk}\Gamma_{jk}^i + W$ considers both the geometrical properties (*i.e.*, $g, \Gamma$) and neural potential fields. It is noteworthy that these two components are computed simultaneously for every temporal states. Fig.3 shows the detailed numerical scheme for the local representation.

## 3 Learning Representations on Manifolds

In this section, we describe an efficient method for learning densities over manifolds. We begin by analyzing the density estimation methods used in previous studies. Let us assume that both the model and target measures admit densities $p_\theta(t, \cdot)d\mathbb{V} = \mu_t^\theta$ and $p_\nu d\mathbb{V} = \nu_t$, respectively. Conventional methods (*e.g.*, RCNF (17)) define the minimization problem of relative entropy $H(\cdot|\cdot)$ between the model probability measure $\mu_t^\theta$ and the target probability measure $\nu_t$, as follows:

$$\min_{\theta \in \Theta} H(\mu_t^\theta|\nu_t) = \min_{\theta \in \Theta} \mathbb{E}\left[ \log \frac{d\mu_t^\theta}{d\nu_t} \right] = \min_{\theta \in \Theta} \mathbb{E}\left[ \log p_\theta - \log p_\nu \right]. \quad (4)$$

To compute the numerical estimation in (4), conventional methods require a preprocessing step (*e.g.*, Gaussian KDE) to obtain an explicit and accurate estimation of the target density $p_\nu$. Owing to this shortcoming, the induced performance is highly dependent on the preprocessing step, which must be correctly tuned for different applications and manifolds. In contrast, the proposed method directly utilizes a set of observed particles sampled from the target object. Specifically, it is constructed

---

[2]This term is also called a "rolling without slipping" to emphasize the intuition (16; 8).

from the $N$ sampled particles $\{y_t^l\}_{1 \le l \le N}$ defined as $\nu_t = \frac{1}{N} \sum_l^N \delta_{y_t^l}$. Consequently, the proposed approach does not require any distributional information to represent the target object.

**Markov Semi-group.** Because our main objective is to find a solution $X_t^\theta$ of the RNSDE, the model measure is defined as a law of the stochastic path $\mu_t^\theta := \mathbb{P}(X_t^\theta \in \cdot) \sim X_t^\theta \in \mathcal{M}$, where $X_t^\theta$ is the solution to the proposed RNSDE. In spite of the rigorous definition, the model measure $\mu_t^\theta$ is not applicable to numerical algorithms owing to its abstract form. To resolve this issue, we suggest an algebraic trick for evaluating the model measure $\mu_t^\theta$. Specifically, we utilize the Markov property of the proposed RNSDE to identify the probabilistic structures of the model measure. Let $P_t^\theta f := \mathbb{E}[f(X_t^\theta)|X_0]$ be a Markov semi-group (28) of $X_t^\theta$. Then, by duality, we can define the probability measure called a *dual semi-group* $P_t^{\theta,*}$ (2), satisfying the following equality:

$$\mathbb{E}_{x \sim \mu_0}\mathbb{E}[f(X_t^\theta)|X_0 = x] = \mathbb{E}_{x \sim \mu_0}[P_t^\theta f(x)] = \mathbb{E}_{x' \sim P_t^{\theta,*}}[f(x')] = \mathbb{E}_{X_t^\theta \sim \mu_t^\theta}[f(X_t^\theta)], \quad (5)$$

where $\mu_0$ indicates the initial state of the RNSDE. Evidently, the expectation according to the measure $\mu_t^\theta$ is identified as the expectation with the dual semi-group $\mu_t^\theta := P_t^{\theta,*}$, which can be easily computed using the relation in (5). Throughout this paper, the model measure is interchangeably denoted as $P_t^{\theta,*}$ and $\mu_t^\theta$.

**Objective Function.** Similar to previous studies, the proposed method aims to minimize the discrepancy between the model and target measures; however, a different approach is adopted. Specifically, we utilize the *static Schrödinger bridge* to define the discrepancy between $P_t^{\theta,*}$ and $\nu_t$:

**Definition 1.** *(Static Schrödinger Bridge (12)) Let us assume that there exists a coupling $\pi_\theta$ such that $\pi_\theta \ll P_t^{\theta,*} \otimes \nu_t$ for fixed time $t \in [0,T]$ where $\pi_\theta \in \Pi(P_t^{\theta,*}, \nu_t)$ is the set of parameterized couplings. Let $R$ be the reference measure satisfying $\frac{dR}{d(P_t^{\theta,*} \otimes \nu_t)} \propto e^{-\gamma^2/2}$ with the Riemannian distance $\gamma$. Then, the (static) Schrödinger bridge $\pi^\star$ is a unique solution to the following minimization problem:*

$$\pi_\theta^\star = \underset{\pi_\theta \in \Pi(P_t^{\theta,*}, \nu_t)}{\arg\min} H(\pi_\theta | R) := \underset{\pi_\theta \in \Pi(P_t^{\theta,*}, \nu_t)}{\arg\min} \int_{\mathcal{M} \times \mathcal{M}} \log\left[\frac{d\pi}{dR}(X_t^\theta, y_t)\right] d\pi(X_t^\theta, y_t). \quad (6)$$

By adopting the Markov property, the proposed Schrödinger bridge problem in (6) enables the transformation of the fixed static bridge suggested in (12) into a temporally variational formulation. In addition, the marginal measure $P_t^{\theta,*}$ of the bridge $\pi_\theta$ is propagated over time by the *parameterized Fokker-Planck Equation* (pFPE) because of the Markov property of $X_t^\theta$. Consequently, the optimal coupling in (6) can be interpreted as being similar to the dynamic Schrödinger bridge explored in a recent study (4) for generative modeling. With the definition of the Schrödinger bridge in (6), our objective function is defined as follows:

$$\min_{\theta \in \Theta} \min_{\pi \in \Pi(P_t^{\theta,*}, \nu_t)} H(\pi_\theta | R) = \min_{\theta \in \Theta} H(\pi_\theta^\star | R). \quad (7)$$

The objective function in (7) first connects the model measure to the target measure by searching the Schrödinger bridge $\pi_\theta^*$ (**inner problem**), and minimizes the relative entropy using the neural network $\theta \in \Theta$ according to $\pi_\theta^*$ (**outer problem**). Unlike the minimization problem suggested in (4), which requires an accurate approximation of the Radon-Nikodym derivative $\frac{d\mu_t^\theta}{d\nu_t}$, the proposed objective function adopts an alternative approach. It is reformulated to identify a feasible scheme for searching for the Schrödinger bridge $\pi_\theta^*$. In the following, we introduce a dual formulation of (7), called the *Markov diffusive Kantorovich dual* to induce the bridge in a computationally tractable manner:

**Definition 2.** *(Markov diffusive Kantorovich dual) Let $\gamma^2$ be the squared Riemannian distance on $\mathcal{M}$. Then, we define the functional $\mathcal{J} : C(\mathcal{M}) \times C(\mathcal{M}) \times \mathbb{R}^+ \times [0,T] \times \Theta \to \mathbb{R}$, as follows:*

$$\mathcal{J}([A,B], \epsilon, t, \theta) = \int A(x) dP_t^{\theta,*}(x) + \int B(y) d\nu_t(y)$$
$$- \epsilon \int e^{A(x)/\epsilon + B(y)/\epsilon - \gamma^2(x,y)/2\epsilon} d(P_t^{\theta,*} \otimes \nu_t). \quad (8)$$

In the following proposition, we relate the proposed dual functional to the regularized version of the Schrödinger bridge problem in (6) in the following proposition:

**Proposition 2.** *(Log-Sinkhorn with Dual formulation) Let $H^\epsilon$ be a regularized relative entropy*[3]. *Then, the Schrödinger bridge problem in* (7) *has a dual formulation in the following equality:*

$$\min_{\theta \in \Theta} \min_{\pi \in \Pi(P_t^{\theta,*}, \nu_t)} H^\epsilon(\pi_\theta | R) = \min_{\theta \in \Theta} \mathcal{J}([A^\star, B^\star], \epsilon, t, \theta) \coloneqq \min_{\theta \in \Theta} \mathcal{J}^*(\epsilon, t, \theta), \quad (9)$$

*where $\epsilon > 0$ is a predetermined constant, $[A^\star, B^\star]$ is the fixed points of log-Sinkhorn operator $\mathcal{S} \coloneqq \mathcal{H}_\mu^\epsilon \circ \mathcal{H}_\nu^\epsilon$, and each functional $\mathcal{H}_\mu^\epsilon, \mathcal{H}_\nu^\epsilon : C(\mathcal{M}) \to C(\mathcal{M})$ is defined as follows:*

$$\begin{cases} \mathcal{H}_\mu^\epsilon[A](y) = \epsilon \log \int_{\mathcal{M}} e^{-\gamma^2(X_t^\theta, y)/\epsilon - A(X_t^\theta)/\epsilon} dP_t^{\theta,*}, \\ \mathcal{H}_\nu^\epsilon[B](x) = \epsilon \log \int_{\mathcal{M}} e^{-\gamma^2(x, y_t)/\epsilon - B(y_t)/\epsilon} \nu_t. \end{cases} \quad (10)$$

Note that $\epsilon = 1$ restores the original problem posed in (7). In (22; 13), they showed that the evaluation of $\mathcal{J}^\star$ on the right-hand side of (9) is the entropic regularized 2-Wasserstein distance. This indicates that the proposed objective function approximates the Wasserstein distance to minimize the discrepancy between model and target measures given a small $\epsilon \approx 0$. The optimal solution to the inner problem is obtained in Proposition 2 by combining the Markov diffusive Kantorovich duality and log-Sinkhorn operator. Our next goal is to solve the outer problem by optimizing neural networks $(\beta, W)$.

**Gradient Descent Scheme.** In the expectations of Definition 2, the Markov dual semi-group $P_t^{\theta,*}$ plays a central role, explicitly revealing the neural networks $\beta$, and $W$ in the equations by reformulating the Kantorovich functional. To specify the discussion, we introduce the equivalent formulation of the first term in (8):

$$\int A(X_t^\theta) dP_t^{\theta,*} \overset{(5)}{=} \int P_t^\theta A(X_0) d\mu_0 = A(X_0) + \int_0^t \mathbb{E}\left[\beta(\theta)\Delta_{\mathcal{M}} A(X_s^\theta) - W_\theta A(X_s^\theta)\right] ds, \quad (11)$$

where $\Delta_{\mathcal{M}}$ is the Laplace-Beltrami operator on $\mathcal{M}$, and $\mu_0$ is the law for the initial state of the RNSDE. Owing to the explicit relation above, the gradient of the first term in (11) can be expressed by calculating the gradients of the neural networks, $\partial_\theta \beta, \partial_\theta W$. Regarding this relation, we propose the gradient flow-based update rule to minimize the dual formulation in (9) with neural networks in the following proposition.

**Proposition 3.** *The following gradient descent scheme $\{\theta_{(m)}\}_{m \in \mathbb{N}^+}$ minimizes the dual functional in the right-hand side of* (9):

$$\theta_{m+1} = \theta_m - \kappa \mathbb{E}\Bigg[ \int_0^{T \wedge \tau} \partial_\theta \beta(\theta_m) g^{ij} \partial_{ij} A^*(X_s^{\theta_m}) ds$$

$$+ \int_0^{T \wedge \tau} \partial_\theta \beta(\theta_m) g^{jk} \Gamma_{jk}^i \partial_i A^*(X_s^{\theta_m}) ds - \int_0^{T \wedge \tau} \partial_\theta w^j(X_s^{\theta_m}; \theta_m) \partial_j A^*(X_s^{\theta_m}) ds \Bigg], \quad (12)$$

*where $\kappa$ is the learning rate, and $\tau \propto c_1$ is the random stopping time that ensures the avoidance of the gradient explosion induced by geometric effects. Then, there exist numerical constants $c_0, c_1, C_0, C_1 > 0$, such that the convergence speed is bounded as $O\left[n^3(\tau \wedge T) C(c_0, c_1, C_0, C_1)\right]$.*

Detailed discussions on the derivation and relation between (8) and (12) are provided in Appendix C.5. Algorithm 1 summarizes the training procedure for the proposed RNSDE.

**Role of Stopping Time $\tau$.** As shown in (12), the proposed gradient descent method requires the evaluation of geometric terms to update the parameters of the RNSDE. In particular, the Riemannian co-metric (*i.e.*, $g^{ij}(X_t^\theta)$) and Christoffel symbol (*i.e.*, $\Gamma_{jk}^i(X_t^\theta)$) are required to be consecutively calculated for the given stochastic trajectory $X_t^\theta$. As addressed in (17), these geometric quantities are not generally bounded. Thus they may cause the gradient explosion problem during the parameter update of the RNSDE. Regarding the issue, the role of stopping time $\tau$ is to ensure safe training by imposing a constraint on the gradient norm. These problems are not problematic in Euclidean geometry (*e.g.*, Euclidean SDE) because the geometric terms are set to constants (*i.e.*, $g^{ij} = 1, \Gamma_{jk}^i = 0$). Thus, an additional module (*e.g.*, $\tau$) is not required in the Euclidean SDE because the raised problem is purely a geometric consequence depending on the non-flat Riemannian structure of manifolds.

---

[3]Please refer to Appendix C.4 for detailed information.

---

**Algorithm 1** Learning Representations on Manifolds with the proposed RNSDE

---

**Require:** Neural parameters $\theta_{m=0} \in \Theta$, learning rate $\kappa$, time interval $\Delta_t$, stopping time threshold $c_1$.

  **for** $m = 1$ to $M$ **do**

    **1)** Simulate the proposed RNSDE by applying the Euler-Maruyama scheme in (14), $\{X_{t_o}^{\theta_m}\}_{1 \leq o \leq T/\Delta_t}$.

    **for** $o = 1$ to $T/\Delta_t$ **do**

      **1-1)** Update the geometric terms according to the current state $(t, X_{t_o}^{\theta_m})$.

$$g^{ij} \leftarrow g^{ij}(X_{t_o}^{\theta_m}), \Gamma_{jk}^i \leftarrow \Gamma_{jk}^i(X_{t_o}^{\theta_m}), \partial_j^{t_o} \leftarrow \partial_j^{t_o}(X_{t_o}^{\theta_m}) \tag{13}$$

      **1-2)** Sample the mean-zero standard Gaussian random variable $\mathbf{Z} \sim \mathcal{N}(0, I_n)$.

      **1-3)** Propagate stochastic trajectories:

$$X_{t_{o+1}}^{\theta_m} = \beta(\theta_m) \sum_j \sqrt{g^{ij}\Delta_t} \mathbf{Z}_j + \left[ -\frac{\beta(\theta_m)}{2} \sum_{j,k} g^{jk}\Gamma_{jk}^i + \sum_j w^j(X_{t_o}^{\theta_m}; \theta_m)\partial_j^{t_o} \right]\Delta_t \tag{14}$$

    **end for**

    **2)** Search the fixed point $[A^\star, B^\star]$ by applying the log-Sinkhorn iteration.

    **3)** Sample random stopping-time $\tau = \inf_{1 \leq t_o \leq T/\Delta_t}\{t_o; |g^{ij}|^2 + |\Gamma_{jk}^i|^2 < c_0\}$.

    **4)** Update the neural network $\theta_{m+1} \leftarrow \theta_m$ by applying the proposed gradient descent in (12) with stopping time $\tau$, and the obtained fixed point $A^\star$.

  **end for**

---

## 4 Comparison to Existing Methods

In conventional Riemannian continuous normalizing flows, model density $p_\theta$ is transited according to the following differential equation:

$$\textbf{RCNF:} \quad \frac{\partial}{\partial t} \log p_\theta(z_t) = -\textbf{div}_\mathcal{M}[V_\theta(z_t, t)], \tag{15}$$

where $z_t$ is the ODE flow on a manifold, and $V_\theta$ is the vector field parameterized by the neural network. As shown in (15), the infinitesimal difference between log-densities is proportional to the divergence of the learnable vector field $V_\theta$, where the spatio-temporal latent information need to be priorly encoded in the neural ODE (*i.e.*, $z_t$). While the probability transition rule in (15) is dependent on the temporal derivative of the neural ODE (*i.e.*, $\partial_t z_t$) by the chain rule, the differential equation in (15) implicitly expresses the probability transition. In the proposed method, the *parameterized Fokker-Planck Equation (pFPE)* obeys the transition rule of model density $X_t^\theta \sim p_\theta$:

$$\textbf{pFPE:} \quad \frac{\partial}{\partial t} p_\theta(t, x) = \beta(\theta)\Delta_\mathcal{M} p_\theta(t, x) - \textbf{div}_\mathcal{M}[p_\theta(t, x)V_\theta(t, x)], \tag{16}$$

where $V_\theta(t, \cdot) := W(t, \cdot; \theta)$ is the neural potential field. Instead of borrowing additional dynamics (*i.e.*, ODE $z_t$ in RCNF) to express the probability transitions of density, the pFPE can explicitly express the infinitesimal evolution of density $p(t, x)$ with neural networks $\beta$ and $W$ for both spatio-temporal variables $(t, x)$ and the solution to the pFPE follows the law of path trajectory $X_t^\theta$.

In (26), the FPE has been numerically simulated by linearization to approximate density $p_t$. In contrast, the proposed method can avoid the computational burden of simulating the pFPE, while considering the dual measure $\mu_t^\theta = p_\theta(t, x)d\mathbb{V} \sim X_t^\theta$ as the law of $X_t^\theta$ using the property of the Markovian semigroup to access the probability at time $t$. In the Lie-group valued generative flow model proposed in (10), it showed that the specific form of vector field $V_\theta(t, \cdot) := \nabla \log(p_\nu(t, \cdot)/p_\theta(t, \cdot)) \in T\mathcal{M}$ can also induce the solution to the FPE. However, its representational power may be limited because the network outputs (*i.e.*, Neural ODE) are implicitly utilized to estimate the model density $p_\theta$. In contrast, the neural networks in the proposed model are directly incorporated into the pFPE as a vector field, which enables rich stochastic representations.

## 5 Related Work

MCNF (15) and RODE (18) defined the ordinary differential equation (ODE) on manifolds and adopted continuous normalizing flows (*i.e.*, RCNF) to express the transition of data representations. In their methods, the stochastic transition was expressed as the ODE flow of log-density on manifolds and directly calculated the geometric operations (*e.g.*, divergence). In a similar context, (10) adopted ODE-based diffeomorphic flows for the Lie structure (*e.g.*, $n$-special unitary group $SU(n)$) that preserves equivariance/invariance. EMSRE (23) proposed a new set of expressive normalizing flows on complex geometries, such as torus and spheres. RCPM (6) directly parameterized the convex

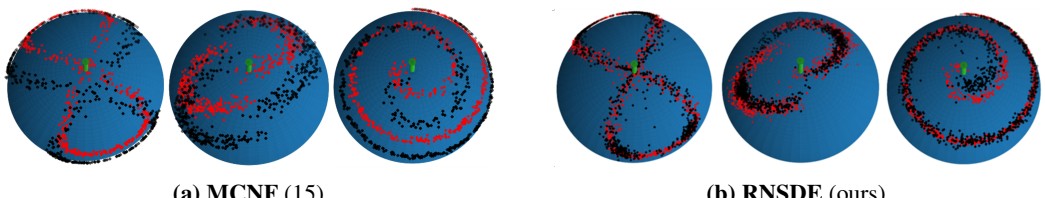

|                     (a) MCNF (15)                     |                     (b) RNSDE (ours)                     |

**Figure 4:** Learned densities on 2-sphere $\mathbb{S}^2$. Green arrows indicate the north poles on the sphere $(0, 0, 1) \in \mathbb{R}^3$.

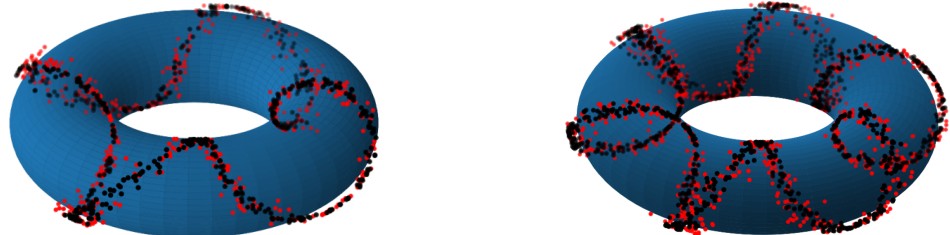

**Figure 5:** Learned densities on 2-Torus $\mathbb{T}^2$. The target shapes are *helix coil* where the number of coils was set to $N_C = 4, 6$ in the left and right figures.

potential map on Riemannian manifolds with a neural network, which can solve the optimal transport problem. MF (24) introduced a computationally tractable linear-type normalizing equation to express the evolution of density over time. In (7; 4), generative modelings was formulated by adopting the Euclidean *dynamic* Schrödinger bridge and SDEs were estimated for applications.

## 6  Experiments

In this section, we compared the proposed RNSDE to baseline methods for various tasks, including *generative modeling, interpolation*, and *reconstruction*. Although the proposed method can be applied to general compact manifolds, we focused on well-known structures. In particular, the following two model manifolds were considered: 2-sphere $\mathbb{S}^2$ and 2-torus $\mathbb{T}^2 \cong \mathbb{S}^1 \times \mathbb{S}^1$. The corresponding Riemannian metrics and detailed information on geometric calculations are provided in Appendix D.

**Generative Modeling.** In this experiment, we evaluated the performance for density estimation on model manifolds of geometric shapes: *8-shapes, two moons, spiral, and helix coil*. Note that the complexities of the target densities in our experiments are considerably higher than those in previous studies (*e.g.*, a mixture of von Mises). Every methods take the initial states of the stochastic trajectories as the mean-zero standard Gaussian in the local coordinate, $X_0^\theta \sim \boldsymbol{\mu}_0 := \psi_\# \mathcal{N}(0, \mathbf{I}_n)$, where the coordinate function is denoted as $\psi$. The goal is then set to accurately restore the target densities. To define the target densities for baseline models, including MCNF (15), RCPM (6), and EMSRE (23), we utilized samples from target densities as anchor points of Gaussian kernels, where the bandwidths were identically set to 10, as suggested in their methods. After obtaining the approximated target densities from the Gaussian KDE, each baseline model was trained by minimizing KL divergence. To train EMSRE, the number of transformation modules and radial components were set to $N_T = 24$ and $K = 5$, respectively. For RNODE, which is a deterministic version of our RNSDE, we utilized only the potential field term in (2)-(A), where the diffusive term was set to 0.

In Tables 1, 2, and 3, we estimated the 2-Wasserstein distance (*i.e.*, $\mathcal{W}_2$) between the empirical target measure $\nu_t$ and the model measure $\mu_t^\theta$ to evaluate the model performance. Table 1 summarizes the performance comparisons of the density estimation tasks. Evidently, the proposed method (RNSDE) outperforms all the baselines by approximately $0.1 \sim 8.2\%$ in all four experiments **(i)**-**(iv)**, despite incorporating no prior information on the target density. RNODE exhibits an inferior performance compared to the proposed RNSDE, which empirically verifies the effectiveness of stochastic representation (*i.e.*, stochastic development) compared to a deterministic one. In the case of arbitrary manifolds with no definite formulas for extrinsic geometric operations (*e.g.*, global exponential map by vector projection from $\mathbb{R}^d$), flow-based models (*e.g.*, MCNF) cannot be implemented in such a geometry because their methods require explicit computation of geometric operations. Accordingly, MCNF is incapable of generating samples on the torus. Unlike their methods, RNSDE only requires intrinsic and local geometric objects (*e.g.*, Riemannian metric tensor,

**Table 1:** Performance comparison of density estimation on synthetic datasets. Each model was evaluated with the 2-Wasserstein distance $\mathcal{W}_2$ ($\times 10^{-2}$). The best performance is rewritten in **bold**.

| Manifold | Density | MCNF | EMSRE | Moser | RCPM | RNODE | RNSDE |
|---|---|---|---|---|---|---|---|
| Sphere | **(i)** 8-shapes | 11.26 | 9.83 | 5.81 | 9.40 | 8.01 | **5.67** |
|  | **(ii)** Two moons | 14.34 | 9.11 | 6.10 | 9.02 | 7.68 | **5.66** |
|  | **(iii)** Spiral | 15.15 | 10.13 | 7.56 | 9.25 | 9.32 | **6.97** |
| Torus | **(iv)** Helix coil | – | – | – | 7.64 | 3.18 | **2.86** |

**Table 2:** Shape interpolation

| Methods | $(i) \to (ii)$ | $(i) \to (iii)$ |
|---|---|---|
| EMSRE | 15.37 | 11.40 |
| RCPM | 9.32 | 9.24 |
| RNODE | 7.09 | 9.01 |
| RNSDE | **6.09** | **7.61** |

**Table 3:** Volcano dataset

| Methods | $\mathcal{W}_2$ |
|---|---|
| EMSRE | 5.70 |
| RCPM | 1.83 |
| RNSDE | **1.08** |

**Table 4:** Vessel Route dataset

| Methods | $\mathbb{E}\gamma^2$ | $\mathcal{W}_2$ |
|---|---|---|
| Latent ODE | 14.25 | 7.17 |
| ODE RNN | 16.37 | 7.29 |
| RNSDE | **12.83** | **6.23** |

Christoffel symbols, and geodesic) to define the stochastic representations; thus, the experimental results can be easily followed.

Figs.4a and 4b show samples from the learned densities (*i.e.*, **black** dots) and target densities (*i.e.*, **red** dots) of the baseline and proposed method, respectively. In the figures, our RNSDE accurately restored the complex target shapes, whereas the MCNF failed to capture the geometric patterns. The samples from the learned densities on 2-Torus are shown in Fig.5, where the number of coils is set to $N_c = 4$ and $N_c = 6$. As shown in the figure, the proposed method correctly approximates the target shapes, regardless of the complexity. These qualitative results highlight the representational power of the proposed RNSDEs for various geometries.

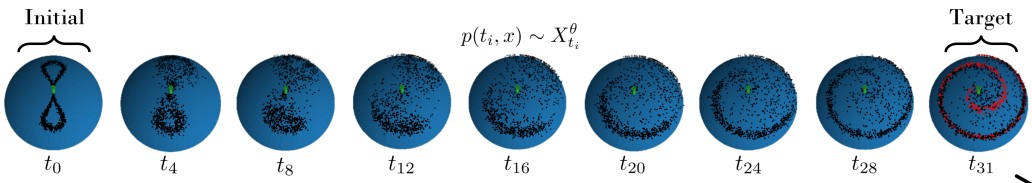

**Figure 6:** Result of shape interpolation starting from *8-shape* to *spiral* on $\mathbb{S}^2$. The **red** (at $t_{31}$) and **black** dots represent target and model densities, respectively.

**Shape Interpolation.** In the second experiment, we conducted shape interpolation starting from **(i)** 8-*shapes* to **(ii)** *two moons* and **(iii)** *spiral*. The total interpolation time ($T$) was set to 0.1, and 32 intermediate samples were taken in the sequence. Fig.6 shows sampled shapes with uniform temporal intervals. Although the distributional constraint is only imposed at the end of the sequence (*e.g.*, target), the intermediate samples exhibit smooth changes between increasing temporal states. This example emphasizes the representability of the proposed RNSDE, showing the superiority of the pFPE, which can express the entanglement of spatio-temporal variables. Table 2 shows the estimated performance comparison on a scale of $10^{-2}$. As shown in the table, the proposed model outperformed baselines by approximately $1.0 \sim 9.3\%$, which is similar to the performance improvement in the previous experiment. Generally, quantitative and qualitative results of density estimation tasks show the versatility of various geometric spaces (*i.e.*, sphere and torus) compared to the previous models. In the following experiment, we evaluated the expressiveness of the proposed model on real-world datasets, including *volcano* and *vessel route*.

**Volcano Eruption.** As the third application, we modeled the density of a volcano eruption dataset (19), which has been explored in previous studies works (10; 17; 24). Table 3 shows the quantitative results of the density estimation on the geographic dataset at a scale of $10^{-1}$. Owing to the dependency on hyperparameters for approximating target densities (*i.e.*, optimal bandwidth for KDE), baseline methods produced relatively poor results in learning the target density, as shown in Table 3. On the contrary, the proposed model precisely restored the intricate distribution of volcano datasets and demonstrated the superiority of expressivity. Fig.7 highlights the samples from the learned (**black**) and target densities (**red**) distributed around the Pacific Rim. In the figure, the proposed model accurately captured the volcano eruptions that occurred on the "Ring of Fire".

**Vessel Route.** In the final experiment, we reconstructed the vessel routes in the ocean, expressed as time sequences. We compared the proposed method with neural ODE models for time-series data, including Latent ODE (5) and ODE-RNN (25). To implement the baseline models in the Riemannian setting, we projected the Euclidean vectors produced by their methods onto the 2-sphere using the stereographic projection. Then, the baseline models were trained to minimize the *mean geodesic errors* (MGEs) (*i.e.*, $\mathbb{E}\gamma^2$). Table 4 shows the comparison results for both MGEs and Wasserstein distances. The results show that RNSDE outperforms other latent models (*e.g.*, Latent ODE and ODE-RNN). Appendix E provides the corresponding figure depicting the sampled trajectories of the trained RNSDE and thoroughly discusses the suggested metric.

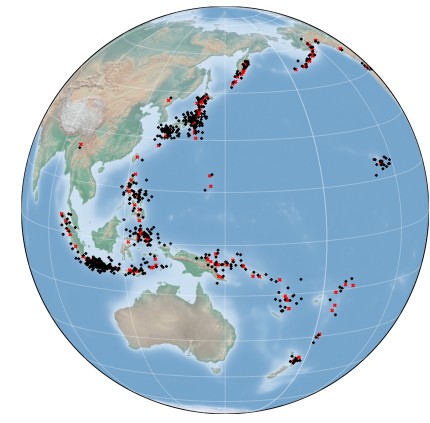

**Figure 7:** Density estimation on the geographic volcano dataset. Table 3 shows the numerical results.

## 7 Discussion

**Conclusion.** In this paper, we presented a principled approach for expressing stochastic representations on manifolds by suggesting a novel Riemannian neural SDE. To define diffusivity, we followed the Eells-Elworthy-Malliavin interpretation, where the Brownian motion on manifolds is derived from the stochastic trajectories on the orthogonal frame bundle. Theoretically, we combined the Markov property with the static Schrödinger bridge problem and proposed a Markov diffusive dual formulation. Then, the corresponding gradient descent scheme was proposed. Empirical results for various tasks including generative modeling, interpolation, and reconstruction, showed that the proposed method surpasses its counterpart.

**Limitation and Future Work.** In the implementation of the proposed RNSDE, we focused on Euclidean embedded manifolds despite their extensive applicability to general compact manifolds. Regarding this issue, we plan to extend our method to other geometries including the non-compact Lie group (*e.g.*, $SU$) for quantum physics (10; 3) and the product spheres for human motion data and global weather data as high-level applications.

**Acknowledgement.** This work was partly supported by the National Research Foundation of Korea (NRF) grant funded by the Korea government (NRF2020R1C1C1004907) and partly supported by Ministry of Culture, Sports and Tourism and Korea Creative Content Agency (Project Number: R2021040032).

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
