# Appendix

**Sung Woo Park**[1], **Hyomin Kim**[2], **Kyungjae Lee**[3], **Junseok Kwon**[4]
[1,2,4]School of Computer Science and Engineering, Chung-Ang University, Korea
[1,3,4]Artificial Intelligence Graduate School, Chung-Ang University, Korea
[1]LG AI Research
[1]pswkiki@gmail.com, [2]icecream126@cau.ac.kr
[3]kyungjae.lee@ai.cau.ac.kr, [4]jskwon@cau.ac.kr

## A   Markov Diffusive Kantorovich Dual Formulation

This section provides basic theoretical details on the log-Sinkhorn operator and its convergence results. Let us define two arbitrary functions $A, B \in C^2(\mathcal{M})/\mathbb{R}$. Then, we start by defining operators defined as follows:

$$\mathcal{H}_\mu^\epsilon : C(\mathcal{M}) \to C(\mathcal{M}), \quad \mathcal{H}_\mu^\epsilon[A](y) = \epsilon \log \int e^{-\gamma^2(x,y)/\epsilon - A(x)/\epsilon} \mu_t(dx), \tag{1}$$

$$\mathcal{H}_\nu^\epsilon : C(\mathcal{M}) \to C(\mathcal{M}), \quad \mathcal{H}_\nu^\epsilon[B](x) = \epsilon \log \int e^{-\gamma^2(x,y)/\epsilon - B(y)/\epsilon} \nu_t(dy). \tag{2}$$

The composition of two operators (*i.e.*, $\mathcal{H}_{\mu_t}^\epsilon, \mathcal{H}_\nu^\epsilon$) is referred to as a *log-Sinkhorn* iteration (operator) $\mathcal{S} \triangleq \mathcal{H}_{\mu_t}^\epsilon \circ \mathcal{H}_\nu^\epsilon$. Let us define $A_l$ as the transformed shape of $A$ after the $l$-th iteration: $A_l = \mathcal{S}^l(A) = \underbrace{\mathcal{S} \circ \mathcal{S} \cdots \circ \mathcal{S}}_{l \text{ time}}(A)$. Then, we define the functional $\overline{\mathcal{F}}(A_l) : C^2(\mathcal{M})/\mathbb{R} \to \mathbb{R}$, as follows:

$$\overline{\mathcal{F}}(A_l) = \int \mathcal{S}^l \circ A(x) d\mu_t^\theta(x) + \int \mathcal{H}_\mu^\epsilon[\mathcal{S}^l \circ A](y) d\nu_t(y). \tag{3}$$

The log-Sinkhorn iteration uniquely minimizes the functional $\overline{\mathcal{F}}$ by the following proposition.

**Proposition 1.** *The log-Sinkhorn iteration $\mathcal{S}$ has the a point in $C^2(\mathcal{M})/\mathbb{R}$. This fixed point is determined up to an additive constant, and minimizes the functional $\overline{\mathcal{F}}$ uniformly:*

$$\overline{\mathcal{F}}(\mathcal{S} \circ A_l) \triangleq \overline{\mathcal{F}}(A_{l+1}) \leq \overline{\mathcal{F}}(A_l). \tag{4}$$

We assume that, for a large enough $l > L$ with small enough $\epsilon \approx 0$, the log-Sinkhorn iteration converges, *i.e.*, $\mathcal{S} \circ A_l = A_l$, and the functional $\overline{\mathcal{F}}$ is minimized. Then, the function $A_{l \vee L}$ is approximated to the $d^2/2$-Legendre transformation (11) of the function $B_{m \vee M}$.

$$[A_{l \vee L}]^c \approx \chi_{\mathbf{supp}(\mu_t)} + B_{l \vee L}, \tag{5}$$

where $[f]^c$ denotes the $d^2/2$-Legendre transformation of $f$ and $\chi_V$ is the support function on the subset $V \subset \mathcal{M}$ defined as:

$$\chi_V(x) = \begin{cases} 0 & \text{if} \quad x \in V \\ \infty & \text{if} \quad x \notin V \end{cases}. \tag{6}$$

Please note that we denote the **fixed point** of $\mathcal{S}$ in the main paper as $A^* := A_{l \vee L}$. The following theorem states that the fixed point of the log-Sinkhorn operator induces optimal transport map:

**Theorem 1.** *(Unique fixed-point of $\mathcal{S}$ and Optimal transport (2)) We assume that $(\mu_t^{(\epsilon)}, \nu_t^{(\epsilon)}) \to (\mu_t, \nu_t)$ in $\mathcal{P}(\mathcal{M})$-weak sense. If $A$ is a fixed point of the log-Sinkhorn operator $\mathcal{S}$ on $C^2(\mathcal{M})/\mathbb{R}$,*

36th Conference on Neural Information Processing Systems (NeurIPS 2022).

*then B converges uniformly to a $d^2/2$-Legendre transformation of A, and it converges to the optimal transport map $\Phi$ satisfying:*

$$\Phi(y) = \exp_y(\nabla_g B(y)), \quad \Phi_\# \nu = \mu_t, \tag{7}$$

*where $\exp_{(.)} : T_{(.)}\mathcal{M} \to \mathcal{M}$ and $\nabla_g$ are Riemannian exponential and gradient, respectively.*

In short, the proposed iteration approximates the $d^2$-convex function, *i.e.*, $B \approx A^c$, and induces the solution to the optimal transport problem (*i.e.*, 2-Wasserstein distance ((8))) as we approximate $\epsilon \to 0$. This fundamental result with the Kantorovich duality theorem and the theoretical characteristic of log-Sinkhorn iteration assure the optimal transport between $\mu_t$ and $\nu_t$.

## B   Parameterized Fokker-Planck Equation on Manifolds

This section provides the omitted information in Section 5 of the main paper. Specifically, the intuition and derivation of (16) in Section 4 of the main paper are shown in the following contents.

**Parameterized Infinitesimal Generator.** We define the *infinitesimal generator* as follows:

$$\mathcal{L}^\theta f = \lim_{t \to 0} \frac{1}{t} P_t^\theta f - f. \tag{8}$$

We define the $P_t^\theta$ (*i.e.*, Markov semi-group) in the main paper. However, because the form in (8) is not applicable due to the $\lim$ operation, we present the alternative differential operator on $C_0^\infty$ as follows:

$$\mathcal{L}_t^\theta f(x) := \beta(\theta)\Delta_\mathcal{M} f(x) + W_\theta f(x), \quad x \in \mathcal{M}, \quad f \in C_0^\infty(\mathcal{M}), \tag{9}$$

where $\Delta_\mathcal{M} := \mathbf{div}_\mathcal{M} \circ \nabla_\mathcal{M}$ is the Laplace-Beltrami operator on $\mathcal{M}$. Notably, due to the time-dependent behavior of neural potential field, the proposed generator induces the time-inhomogeneous Markov process. Then, the *Fokker-Planck equation* or *Kolmogorov forward equation* is defined as follows:

$$\partial_t p_\theta(t, x) = [\mathcal{L}_t^\theta]^* p_\theta(t, x) = \Delta_\mathcal{M} p_\theta(t, x) - \mathbf{div}\,[p_\theta(t, x)W_\theta], \tag{10}$$

where $[\mathcal{L}_t^\theta]^*$ is the adjoint differential operator of $\mathcal{L}_t^\theta$ satisfying the equality in $\mathbb{L}_2(d\mathbb{Q})$ sense:

$$\int_\mathcal{M} p_\theta(t, x)[\mathcal{L}_t^\theta]^* q(t, x)d\mathbb{V} = \int_\mathcal{M} p_\theta(t, x)\mathcal{L}_t^\theta q(t, x)d\mathbb{V} \quad \forall t \geq 0 \tag{11}$$

for arbitrary density $q$ on $\mathcal{M}$.

## C   Proofs

### C.1   Assumptions

This section enumerates the technical assumptions used in the proof.

- **(H1)** Any Kantorovich potential function $A \in C_0^\infty(\mathcal{M})$ in the contents is $\mu_t^\theta \times \mathbb{L}([0, T])$ measurable for Lebesgue measure $\mathbb{L}$ on the time interval $[0, T]$.

- **(H2)** Any Kantorovich potential function $A$ in the contents satisfies the inequality: $|\mathcal{L}^\theta A| < \infty$ almost everywhere $\mathbb{P}$, for all $\theta \in \Theta$.

- **(H3)** The parameter space $\Theta$ is a compact subset of $\Theta \subset \mathbb{R}^{D'}$.

- **(H4)** The Kantorovich potentials $[A, B]$ have bounded norm:

$$\mathbb{E}_{\mu_t^\theta}\left[ \|\partial_i A\|_E^2 + \|\partial_{ij} A\|_E^2 + \|\partial_i B\|_E^2 + \|\partial_{ij} B\|_E^2 \right] \leq c_0. \tag{12}$$

- **(H5)** Constants $C_0^\theta, C_1^\theta$ exist such that $\sup_{\theta \in \Theta} \|\partial_\theta \beta(\theta)\|_E^2 \leq C_0^\theta$, $\sup_{\theta \in \Theta} \|\partial_\theta w^j\|_E^2 \leq C_1^\theta$, a.e, $[\mu_t^\theta]$.

## C.2 Lemmas and Definitions

In this section, we provide lemmas and definitions considered in the proofs.

**Definition 1.** *(Legendre transformation) The c-Legendre transformation is the function $A^c \in C(\mathcal{M})$ for $A \in C(\mathcal{M})$*

$$A^c(y) := \sup_{x \in \mathcal{M}} \left( -c(x, y) - A(x) \right), \tag{13}$$

where we follow the sign convention suggested in (2) for Legendre transformation.

**Lemma 1.** *The solution to the proposed RNSDE $X_t^\theta$ is a semi-martingale Markov process.*

*Proof.* In C.3, a semi-martingale property is presented. Then, the proof is completed by showing the Markov property because the second term in the definition of RNSDE is a deterministic process.

$$\mathbb{E}[\beta(\theta)\pi^{-1}(X_t^\theta) \circ dB_t | \mathcal{F}_s] = \mathbb{E}[\beta(\theta)U_t^\theta \circ dB_t | X_t^\theta] = 0. \tag{14}$$

The equality is trivial where the conditional expectation is taken in the orthonormal frame bundle. $\square$

**Lemma 2.** *There exists one-to-one correspondences between Itó and Stratonovich SDEs:*

$$X_t \circ dY_t = Y_t dZ_t + \frac{1}{2}[X, Y], \tag{15}$$

*where $[Y, Z]$ is the covariation between the stochastic processes $Y$ and $Z$.*

**Lemma 3.** *(6) The following quantities are equivalent:*

$$\Delta_{\mathcal{M}} f := \frac{1}{\sqrt{|g|}} \partial_i \left( \sqrt{|g|} g^{ij} \partial_j f \right) = g^{ij} \partial_{ij} f + g^{jk} \Gamma_{jk}^i \partial_i f. \tag{16}$$

**Lemma 4.** *Given an arbitrary starting point $\mathbf{A}_{l=0}^t(\cdot; \theta) \in C^2(\mathcal{M})/\mathbb{R}$, $\bar{\mathcal{F}}(\mathbf{A}_l^t(\cdot; \theta))$ converges uniformly to $\bar{\mathcal{F}}(\mathbf{A}_{l \vee L}^t(\cdot; \theta))$.*

*Proof.* The proof directly follows from Theorem 1 and the continuity of $\bar{\mathcal{F}}(\cdot)$ on $C^2(\mathcal{M})/\mathbb{R}$. $\square$

The uniform convergence property plays a central role in the proof of Proposition 3 to combine the log-Sinkhorn iteration with the gradient-flow scheme.

## C.3 Proof of Proposition 1.

In this section, we aim to show the following equivalent relations between various representations of identical objects, **RNSDE**.

$$\underbrace{dU_t^\theta}_{\text{local}} \xleftrightarrow{\varphi} \underbrace{dU_t^\theta}_{(19)} \xleftrightarrow[\pi]{\pi^{-1}} \underbrace{dX_t^\theta}_{(18)} \xleftrightarrow{\varphi} \underbrace{dX_t^\theta}_{\text{local}}. \tag{17}$$

Specifically, the goal of this proof is to represent the RNSDE on $\mathcal{M}$ in a local coordinate (RHS in (17)) by rewriting the local coordinate of the RNSDE on $O\mathcal{M}$ (LHS in (17)).

**(A) Relation between $X_t$ and $U_t$.** To fully represent the proposed RNSDE in a local coordinate system, we clarify the first term (*i.e.*, stochastic development) in the definition of the RNSDE.

$$dX_t^\theta = W(t, X_t^\theta; \theta)dt + \beta(\theta)\pi^{-1}(X_t^\theta) \circ dB_t, \quad \{X_t^\theta\}_{0 \le t \le T} \in \mathcal{M}. \tag{18}$$

In (1; 12), the Brownian motion on $O\mathcal{M}$ is defined as follows:

$$dU_t^\theta = \mathbb{H}_{W_\theta}(U_t^\theta)dt + \beta(\theta) \sum_{i=1}^n \mathbb{H}_{e_i}(U_t^\theta) \circ dB_t, \quad \{U_t^\theta\}_{0 \le t \le T} \in O\mathcal{M}, \tag{19}$$

where we denote $W_\theta(\cdot) := W(t, \cdot; \theta)$ for simplicity. The SDE in (19) on the orthogonal frame bundle is the lift of SDE in (18) on the target manifold where $\mathbb{H}_{W_\theta}$ is the *horizontal vector field* of $W_\theta$, and $\{\mathbb{H}_{e_i}\}_{1 \le i \le n}$ is called a *fundamental horizontal vector field*, which has the following local expression:

$$\mathbb{H}_{e_i}(U_t^\theta) = e_i^j \partial_i - e_i^j e_m^l \Gamma_{jl}^k(X_t^\theta) \bar{\partial}_m^k. \tag{20}$$

While the set of basis vectors $\{\partial_j \triangleq \frac{\partial}{\partial x_j}, \bar{\partial}_m^k \triangleq \frac{\partial}{\partial e_m^k}; 1 \leq j, k, m \leq d\}$ span the tangent of orthogonal fame bundle $TO\mathcal{M}$, $\mathbb{H}_{e_i}$ is the vector field on $TO\mathcal{M}$ expressed as a linear combination of basis vectors $\{\partial_j, \bar{\partial}_m^k\}$.

**(B) Local coordinate on frame bundle.** We denote a generic point of orthogonal frame bundle $O\mathcal{M}$ by $U = (\{X^i\}_{1\leq i\leq n}, \{e_j^i\}_{1\leq i,j\leq n})$ where a frame is the element of orthogonal group $e_j^i \in O(n)$. We have $U e_i = e_i^j \partial_i$ where $\{e_i\}$ is the basis of the Euclidean space $\mathbb{R}^n$. Then, the local expression of (19) is expressed as

$$dU_t^\theta = \begin{pmatrix} dX_t^\theta \\ de_j^i \end{pmatrix} = \begin{pmatrix} e_j^i(t) \circ dB_t^j + W_\theta dt \\ -\Gamma_{kl}^i(X_t^\theta) e_j^l(t) e_m^k(t) \circ dB_t^m \end{pmatrix}. \tag{21}$$

By applying the Itó's lemma to the SDE with the function $f = R \circ \pi$ for an arbitrary $R : \mathcal{M} \to \mathbb{R}$, we can obtain the following representation of the SDE.

$$f(U_t^\theta) = f(U_0) + \int \mathbb{H}_{e_i} f(U_t^\theta) \circ dB_t^i + \int \mathbb{H}_{W_\theta} f(U_t^\theta) dt. \tag{22}$$

In the local coordinate presentation, we can write (22), as follows:

$$\begin{aligned}
dR(X_t^\theta) &= \mathbb{H}_{e_i} R(X_t^\theta) \circ dB_t^i + W(t, X_t^\theta; \theta) R(X_t^\theta) dt \\
&= \mathbb{H}_{e_i} R(X_t^\theta) dB_t^i + \frac{1}{2} d[\mathbb{H}_{e_i} R, B]_t + \mathbb{W}_t^\theta \\
&= R(X_0) + \mathbb{M}_t + \frac{1}{2} \mathbb{N}_t + \mathbb{W}_t^\theta.
\end{aligned} \tag{23}$$

In the second equality, we transform the Stratonovich SDE into Itó's SDE by applying Lemma 2. Because the lifted function $f = R \circ \pi$ uniquely determines the orthonormal basis of tangent space $u e_i \in T_\mathcal{M}$ by the property of the fundamental horizontal vector field $\mathbb{H}_{e_i}$ (*i.e.*, there exists a unique relation $\pi_\star \mathbb{H}_{e_i}(U_t) = U_t e_i$), we can express $\mathbb{H}_{e_i} f(U_t) = U_t e_i$. This fact leads to the second equality in the following formulation:

$$\mathbb{M}_t = \sum_i \mathbb{H}_{e_i} f(U_t^\theta) dB_t^i = \sum_i (df)_{U_t^\theta} [\mathbb{H}_{e_i}(U_t^\theta)] dB_t^i, \tag{24}$$

where $df$ is the differential of $f$. To understand the above relation precisely, we present the local coordinate expression of $\mathbb{M}_t$. Specifically, the horizontal curve $U_t^\theta$ in orthonormal frame bundle $O\mathcal{M}$ can be expressed in a local coordinate, $U_t^\theta = [[X_t^\theta]^i, e_j^i(t)]$, as follows:

$$\begin{aligned}
\mathbb{H}_{e_i}(U_t^\theta)[f] &\triangleq \mathbb{H}_{e_i} f(U_t^\theta) = (df)_{U_t^\theta} [\mathbb{H}_{e_i}(U_t^\theta)] \\
&= e_i^j \partial_j f(U_t^\theta) - e_i^j e_m^l \Gamma_{jl}^k(X_t^\theta) \bar{\partial}_m^k f(U_t^\theta) \\
&= e_i^j \partial_j R \circ \pi([[X_t^\theta]^i, e_j^i(t)]) \\
&= (e_i^j \partial_j)_{X_t} R(X_t^i) = (U_t^\theta e_i)_{X_t^\theta} R([X_t^\theta]^i) \\
&= T([X_t^\theta]^i) R([X_t^\theta]^i).
\end{aligned} \tag{25}$$

The third equality holds as $\bar{\partial}_m^k \circ f([X_t^i, e_j^i(t)]) = \bar{\partial}_m^k R \circ \pi([X_t^i, e_j^i(t)]) = \bar{\partial}_m^k R(X_t) = 0$. In the last equality, we define the vector field $T \in T\mathcal{M}$ as $U_t^\theta e_i = e_i^j \partial_j \triangleq T(X_t)$. To estimate the vector field $T(X_t)$, we need to find out the explicit numeric of the orthogonal matrix $e_i^j$ by solving the following equation:

$$(U_t^\theta e_i)|_{X_t^\theta} = e_i^j \partial_j|_{X_t^\theta}. \tag{26}$$

To solve the equation, we take the Riemannian inner product between $U_t^\theta e_l$ and $U_t^\theta e_m$, as follows:

$$\begin{aligned}
\langle U_t^\theta e_l, U_t^\theta e_m \rangle_{g(X_t^\theta)} &= \langle e_l^i \partial_i, e_m^j \partial_j \rangle_{g(X_t^\theta)} = e_l^i \langle \partial_i, \partial_j \rangle_{g(X_t^\theta)} e_m^j \\
&= e_l^i g_{ij}(X_t^\theta) e_m^j = \delta_{ij},
\end{aligned} \tag{27}$$

where we denote $\langle X, Y \rangle_g$ because the inner product between vector fields $X, Y \in T\mathcal{M}$ and $\delta_{lm}$ is the coordinate delta function. Using the relation in (27), the following identity can be easily obtained:

$$\sum_k e_k^i e_k^j = g^{ij}. \tag{28}$$

We can express the identity in (28) as a matrix form, as follows:

$$E(i,k)^T E(j,k) = G^{-1}(i,j), \tag{29}$$

where $E(i,k) := \{e_k^i\}$ and $G^{-1}(i,j) := g^{ij}$. To obtain the explicit form of the matrix $E$, we apply the Cholesky decomposition to the co-metric matrix. (*i.e.*, $E = \mathbf{Ch} \circ [G^{-1}]$). Finally, the derivation form of horizontal vector field to $f$, $\mathbb{H}_{e_i} f(U_t)$, can be written in the local coordinate as follows:

$$\mathbb{H}_{e_i} f(U_t^\theta) = (U_t^\theta e_i)|_{X_t^\theta} R(X_t^i) = \mathbf{Ch} \circ [G^{-1}(X_t^\theta)]^i \partial_i R([X_t^\theta]^i). \tag{30}$$

Because the co-metric matrix is semi-definite positive regarding the Riemannian structure, the following relation holds by the elementary algebraic property of Cholesky decomposition regarding the SDP matrix:

$$\mathbf{Ch} \circ G^{-1} = G^{-\frac{1}{2}}. \tag{31}$$

Finally, the local martingale term $\mathbb{M}_t$ is written in the local coordinate as follows:

$$\mathbb{M}_t = \sum_i G(X_t^\theta)^{-\frac{1}{2}} \partial_i R(X_t^\theta) dB_t^i. \tag{32}$$

To determine whether the process $\mathbb{M}_t$ is the local martingale, we estimate the quadratic variation of the process $\mathbb{M}_t$ using the following calculation.

$$d\left[\mathbb{M}, \mathbb{M}\right]_t = \left[\sum_i \sqrt{g_{ii}^{-1}} \partial_i R(X_t^\theta)\right]^2 dt. \tag{33}$$

Because the quadratic variation of Euclidean Brownian motion $B$ can be derived as $[B,B]_t = dt$, $\mathbb{M}_t$ is the local martingale.

$$\begin{aligned}
\mathbb{N}_t = [\mathbb{H}_{e_i} R, B]_t &= \int \mathbb{H}_{e_j} \mathbb{H}_{e_i} f(U_t^\theta) d\left[B^j, B^i\right]_t \\
&= \int \sum_i \mathbb{H}_{e_i}^2 f(U_t^\theta) dt = \int \Delta_{\mathcal{OM}} f(U_t^\theta) = \int \Delta_{\mathcal{M}} R(X_t^\theta).
\end{aligned} \tag{34}$$

The last term $\mathbb{W}_t^\theta = W(t, X_t; \theta) R(X_t)$ corresponds to the anti-development of vector field $\mathbb{H}_{W_\theta}$. By collecting the defined stochastic representations, $\mathbb{M}, \mathbb{N}$, and $\mathbb{W}$, the proposed local coordinate expression shown in the main paper is derived. Moreover, this shows that the lift of the solution to the propsoed RNSDE is a semi-martingale process defined on $\mathcal{OM}$.

### C.4 Proof of Proposition 2.

We assume that the reference measure $R$ admits the density: $d(\mu_t^\theta \otimes \nu_t)/dR = e^{\frac{\gamma^2}{2} + \log Z}$, where $Z = \int e^{-\frac{\gamma^2}{2}} d(\mu_t^\theta \otimes \nu_t)$ is the normalizing constant. Then, the regularized (scaled) entropy is defined as the following form:

$$H^\epsilon(\pi_\theta | R) = \frac{1}{2} \int \gamma^2 d\pi_\theta + \epsilon H(\pi_\theta | \mu_t^\theta \otimes \nu_t) + \epsilon \log Z, \tag{35}$$

where $H^\epsilon|_{\epsilon=1} \equiv H$ restores the original relative entropy. Then, the proof is completed by showing the following relation:

$$\lim_{l \to \infty} \bar{\mathcal{F}}(\mathbf{A}_l^t) = J([A^*, B^*], \epsilon, t, \cdot) = \lim_{\epsilon \to 0} H^\epsilon(\pi_\theta^* | R) = \mathcal{W}_2(\mu_t^\theta, \nu_t), \tag{36}$$

where $\pi^*$ is the static Schrödinger bridge. If we apply the log-Sinkhorn iteration $l \to \infty$, then the function $A_l$ converges to the Legendre transform of $B$ (*i.e.*, $\lim_{l \to \infty} A_l \to B^c + \chi_{\mathbf{supp}(\mu_t^\theta)}$) by Theorem 1. Then, we introduce the important equality:

$$\int e^{A^*/\epsilon \oplus B^*/\epsilon - \gamma^2/2\epsilon} d(\mu_t^\theta \otimes \nu_t) = 1, \tag{37}$$

where the equality holds because the function $A^* = [B^*]^c$ is the Legendre transform induced by Theorem 1. This leads to the fact that the third term in the definition of Markov diffusive Kantorovich

dual can be ignored in the minimization problem (*i.e.*, $\partial_\theta \epsilon = 0$) for arbitrary $0 \leq \epsilon < \infty$. Regarding the definition of regularized entropy, the second equality naturally follows. Finally, the last one can be directly obtained from the Kantorovich duality.

**Note.** For a large enough $l \gg L$, the equality in (36) shows that solving the minimization problem $\min_\theta \mathcal{J}$ is equivalent to $\min_\theta \bar{\mathcal{F}}$. In other words, the calculation for the gradient of the third term of functional $\mathcal{J}$ is redundant for the gradient descent scheme in the case of the Kantorovich potentials $[A^*, B^*]$. Moreover, the small value $\epsilon \approx 0$ in our implementation makes the third term negligible. Regarding the discussions, we apply the the gradient descent scheme for the proposed functional $\bar{F}$ rather $\mathcal{J}$ to train the RNSDE.

**Horizontal Diffusion.** Identically, the infinitesimal generator can be defined on the frame bundle as the following formulation:

$$\mathcal{L}_{O\mathcal{M}}^\theta \tilde{f}(u) \coloneqq \beta(\theta)\Delta_{O\mathcal{M}}\tilde{f}(u) + H_{W_\theta}\tilde{f}(u), \quad u \in O\mathcal{M}, \quad \tilde{f} \in C_0^\infty(O\mathcal{M}), \tag{38}$$

where $\Delta_{O\mathcal{M}} = \sum_i^n H_{e_i}^2$ is the famous Bochner's horizontal Laplacian and $H_{e_i}$ is a fundamental horizontal vector field defined in the previous section.

### C.5    Proof of Proposition 3.

**Diffusive Kantorovich functional with dual Semi-group.** Because the proposed process $X_t^\theta$ is the Markov diffusion process regarding the definition of generator in (9) and Lemma 1, we can apply the geometric version (on orthonormal frame bundle) of the Itó's Lemma (12) to obtain the following equality:

$$\mathbf{A}_l^t(X_t^\theta; \theta) = A_l(X_0) + \int_0^t \mathcal{L}^\theta A_l(X_s^\theta)ds + \frac{\beta(\theta)}{2}\int_0^t \langle U_s^{-1}\nabla_g A_l(X_s), dB_s\rangle, \tag{39}$$

where $U_s^{-1} : T_{X_s^\theta}\mathcal{M} \to \mathbb{R}^d$ is the inverse of frame $U_s$ at $X_s$ which is also the solution to the horizontal diffusion process in the main paper. Please note that the notation for $A_l$ is rewritten as $\mathbf{A}_l^t \in [0, T] \times \mathbb{N}^+ \times \mathcal{M} \times \Theta$ in the LHS to emphasize the parameterization in the RHS.

$$\begin{aligned}
\overline{\mathcal{F}}(\mathbf{A}_l^t(\cdot, \theta)) &= \underbrace{\int \mathbf{A}_l^t(X_t^\theta, \theta)dP_t^{\theta,*}}_{\text{Parameter Activated}} + \underbrace{\int \mathcal{H}_\mu^\epsilon[A_l](y)d\nu(y)}_{\text{Parameter Frozen}} \\
&= \mathbb{E}_{x_0 \sim p_0}\mathbb{E}[\mathbf{A}_m^t(X_t^\theta, \theta)|X_0 = x] + \int \mathcal{H}_\mu^\epsilon[A_l](y)d\nu(y) \\
&= \mathbb{E}_{x_0 \sim p_0}\mathbb{E}\left[A_l(x) + \int_0^t \mathcal{L}^\theta A_l(X_s^\theta)ds|X_0 = x\right] + \int \mathcal{H}_\mu^\epsilon[A_l](y)d\nu(y) \\
&= \mathbb{E}_{x_0 \sim p_0}\mathbb{E}\left[A_l(x) + \int_0^t \beta(\theta)\Delta_\mathcal{M} A_l - W_\theta A_l(X_s^\theta)ds|X_0 = x\right] + \int \mathcal{H}_\mu^\epsilon[A_l](y)d\nu(y).
\end{aligned} \tag{40}$$

In the first equality, the parameter in the second term $\int \mathcal{H}_\mu^\epsilon[A_l](y)d\nu(y)$ is considered fixed during the update. In other words, our setting induces the following property:

$$\partial_\theta \left[\int \mathcal{H}_\mu^\epsilon[A_l](y)d\nu(y)\right] = \mathbf{0}_{1 \times D'}. \tag{41}$$

This trick enables us to avoid the calculation of the second term $\mathbb{E}\mathcal{H}_\mu^\epsilon$, and makes our discussion simpler. The second the equality in (40) reveals the proposed Markov interpretation on Kantorovich functional, where the expectation that is taken over $P_t^{\theta,*}$ is replaced by the conditional expectation using the property of the Markov semi-group. The third and fourth equalities are induced by the geometric Ito's formula in (39) and the definition $\mathcal{L}^\theta$. Please note that the expectation over the third term in (39) vanishes by the property of martingale:

$$\mathbb{E}_{\mu_t^\theta}\left[\frac{\beta(\theta)}{2}\int_0^t \langle U_s^{-1}\nabla_g A_l(X_s^\theta), dB_s\rangle\right] = \frac{\beta(\theta)}{2}\mathbb{E}_{\mu_t^\theta}\left[\int_0^t U_s^{-1}g^{ij}dA_l(X_s^\theta)[dB_s]^T\right] = 0. \tag{42}$$

This can be easily shown by the fact that $U_s^{-1}g^{ij}dA_l$ is the deterministic; thus, stochastic integral in (42) is purely the $\mathcal{F}_t$-martingale.

**Gradient flow with respect to neural parameter $\theta \in \Theta$.** To sum up, the gradient of Kantorovich functional $\overline{\mathcal{F}}$ can be rewritten in the dual formulation regarding the equalities in (40):

$$\partial_\theta \overline{\mathcal{F}}(\mathbf{A}_l^t(\cdot, \theta)) = \partial_\theta \mathbb{E} \int_0^t \mathcal{L}^\theta A_l ds = \mathbb{E} \int_0^t \partial_\theta \beta(\theta) \Delta_{\mathcal{M}} A_l ds + \mathbb{E} \int_0^t \partial_\theta W_\theta A_l ds, \quad (43)$$

where the second equality can be obtained by applying the dominated convergence theorem with the assumptions **(H1)** and **(H2)**. In local coordinate, this evaluation can be written as follows:

$$\lim_{l \to \infty} \partial_\theta \overline{\mathcal{F}}(\mathbf{A}_l(X_t^\theta; \theta)) = \partial_\theta \lim_{l \to \infty} \overline{\mathcal{F}}(\mathbf{A}_l(X_t^\theta; \theta)) = \mathbb{E}_{\mu_t^\theta} \left[ \int_0^t \partial_\theta \beta(\theta) g^{ij}(X_t^\theta) \partial_{ij} A_{l \to \infty}(X_s^\theta) ds \right.$$
$$\left. + \int_0^t \partial_\theta \beta(\theta) g^{jk}(X_t^\theta) \Gamma_{jk}^i(X_t^\theta) \partial_i A_{l \to \infty}(X_s^\theta) ds + \int_0^t \partial_\theta w^j(X_s^\theta; \theta) \partial_j A_{l \to \infty}(X_s^\theta) ds \right]. \quad (44)$$

The interchange between gradients and the limitation in the first equality follows the uniform convergence property of Kantorovich functional developed in Lemma 4. The expectation formula can be derived by calculating the evaluation in (41) with the fact $\partial_\theta A_l = 0$ a.e, $[\mathbb{P}]$. Then, we introduce the gradient flow to minimize the functional $\overline{\mathcal{F}}$ defined as follows:

$$d\theta(s, l) = -\kappa \partial_\theta \overline{\mathcal{F}}(\mathbf{A}_l^t(X_t^\theta; \theta))|_{\theta=\theta(s)}, \quad (45)$$

where the auxiliary variable $s$ denotes the iteration of gradient flows with respect to the neural parameter. By taking limitations to both variables $s$ and $l$, we can obtain the following evaluation:

$$\lim_{l,s \to \infty} d\theta(s, l) = -\kappa \partial_\theta \lim_{l \to \infty} \bar{\mathcal{F}}(\mathbf{A}_l^t(X_t^\theta, \theta))|_{\theta(s \to \infty)}. \quad (46)$$

The equality $\partial_\theta \lim_l = \lim_l \partial_\theta$ can be obtained by Lemma 4.

$$\arg\min_{\theta \in \Theta} \min_{C(\mathcal{M})/\mathbb{R}} \bar{\mathcal{F}}(\mathbf{A}^t(\cdot, \theta))|_\theta = \lim_{s \to \infty} \lim_{l \to \infty} d\theta(s, l). \quad (47)$$

While the sequence $d\theta(s, \infty)$ uniquely determines the equality $\min_{C(\mathcal{M})/\mathbb{R}} \bar{\mathcal{F}}(\mathbf{A}^t(\cdot, \theta))|_\theta = \bar{\mathcal{F}}(\mathbf{A}_\infty^t(\cdot, \theta))|_\theta$ for every $s \in \mathbb{N}^+$, the proposed gradient flow in (46) is well-defined and minimizes the functional $\bar{\mathcal{F}}$. In Algorithm 1 of the main paper, the numerical procedure of the proposed gradient flow is presented.

**Gradient Explosion according to Geometric effect.** As shown in (44), the proposed gradient flow requires the evaluation of the geometric effect to update the parameters of the RNSDE. In particular, Riemannian co-metric (*i.e.*, $g^{ij}(X_t^\theta)$) and the Christoffel symbol (*i.e.*, $\Gamma_{jk}^i(X_t^\theta)$) are consecutively calculated for the given stochastic trajectory $X_t^\theta$. Unfortunately, these geometric evaluations are not generally bounded, which may cause the gradient explosion problem during the parameter update of the RNSDE. The following example reveals the aforementioned problem:

**Example: Gradient Explosion on $\mathbb{S}^2$.** We denote the local coordinate of 2-sphere as $X_t^\theta = [\vartheta_t^\theta, \varphi_t^\theta]$. Then, we can induce the relation:

$$\sup_{i,j} \left[ |g^{ij}(X_t^\theta)| + |\Gamma_{jk}^i(X_t^\theta)| \right] = 2|\csc^2(\vartheta_t^\theta)| \vee 1, \quad (48)$$

where the equality in (48) holds by the fact $\csc^2(\vartheta_t^\theta) - \cot(\vartheta_t^\theta) \geq 0$, a.e, $[\mathbb{P}]$. Subsequently, we define the union of metric balls centered at $\{m\pi\}_{m \in \mathbb{N}^+}$ as follows:

$$\bigcup_{m \in \mathbb{N}^+} B(m\pi, \epsilon) \times \{\varphi_t^\theta\} \subset \mathcal{M}. \quad (49)$$

If we take small radii $\epsilon \to 0$ of metric balls, the upper bound of (48) diverges:

$$\lim_{\epsilon \to 0} \left[ \bigcup_{m \in \mathbb{N}^+} B(m\pi, \epsilon) \times \{\varphi_t^\theta\} \right] \iff \csc^2(\vartheta_t^\theta) \uparrow \infty. \quad (50)$$

In this case, the gradient flow defined in (46) is not well-defined due to the divergence of geometric terms. To tackle the problem, we propose random stopping time to avoid an undesirable result:

$$\tau = \inf_{t \in [0,T]} \left\{ t; \sup_{1 \leq i,j \leq n} |g^{ij}(X_t^\theta)|^2 + |\Gamma_{jk}^i(X_t^\theta)|^2 > c_1 \right\}. \quad (51)$$

In (51), the stopping time $\tau$ defines the threshold $c_1$ to bound the geometric effect. While the stopping time $\tau$ of $X_t^\theta$ induces the identical stopping time $\tau$ of horizontal lift $U_t^\theta$ (Lemma 2.3.7 (4)), the proposed RNSDE defined on the frame bundle is also controlled. We determine whether the convergence speed of gradient flow is bounded by specifying the following evaluation:

$$\frac{\partial}{\partial s}\bar{\mathcal{F}}(\mathbf{A}_l^{t\wedge\tau}(X_{t\wedge\tau}^\theta;\theta))|_{\theta=\theta(s)} = -\kappa\left\|\partial_\theta\lim_{l\to\infty}\overline{\mathcal{F}}(\mathbf{A}_l^{t\wedge\tau}(X_{t\wedge\tau};\theta))|_{\theta=\theta(s)}\right\|_E^2, \qquad (52)$$

where the RHS in (52) can be calculated as follows:

$$\left\|\partial_\theta\lim_{l\to\infty}\overline{\mathcal{F}}(\mathbf{A}_l^{t\wedge\tau}(X_{t\wedge\tau}^\theta;\theta))\right\|_E^2 \leq \mathbb{E}_{\mu_{t\wedge\tau}^\theta}\left[\|\partial_\theta\beta(\theta)\|_E^2\int_0^{t\wedge\tau}\sum_{i,j}|g^{ij}(X_t^\theta)|^2\,|\partial_{ij}A_{l\to\infty}|^2\,ds\right]$$

$$+ \mathbb{E}_{\mu_{t\wedge\tau}^\theta}\left[\|\partial_\theta\beta(\theta)\|_E^2\int_0^{t\wedge\tau}\sum_i\sum_{j,k}\left|g^{jk}(X_t^\theta)\right|^2\left|\Gamma_{jk}^i(X_t^\theta)\right|^2\left|\partial_iA_{l\to\infty}(X_s^\theta)\right|^2\,ds\right]$$

$$+ \mathbb{E}_{\mu_{t\wedge\tau}^\theta}\left[\int_0^{t\wedge\tau}\sum_j\left\|\partial_\theta w^j(X_s^\theta;\theta)\right\|_E^2\left|\partial_jA_{l\to\infty}(X_s^\theta)\right|^2\,ds\right]$$

$$\leq \frac{n}{2}(n+1)(t\wedge\tau)c_0c_1C_0^\theta(1+n) + \frac{n}{2}n(n+1)(t\wedge\tau)c_0^2c_1C_0^\theta + n(t\wedge\tau)C_1^\theta c_0$$

$$\approx O\left(n^3(\tau\wedge T)C(c_0,c_1,C_0^\theta,C_1^\theta)\right). \quad (53)$$

The first inequality follows by applying Jensen's inequality. While the Kantorovich potential $[A^*]^c$ is the $(d^2/2)$-Legendre transform of $B^*$, we can obtain the equalities in (54) and (55).

$$\left|\lim_{l\to\infty}\partial_iA_l\right|_E^2 = \left|\partial_i[\lim_{l\to\infty}A_l]^c\right|^2 = |\partial_i[A^*]^c|^2 = |\partial_iB^*|^2, \quad \forall 1\leq i\leq n. \qquad (54)$$

Similarly, the second derivatives of the potential function converge to the following evaluation.

$$\left|\lim_{l\to\infty}\partial_{ij}A_l\right|^2 = \left|\partial_{ij}[\lim_{l\to\infty}A_l]^c\right|^2 = |\partial_{ij}[A^*]^c|^2 = |\partial_{ij}B^*|^2, \quad \forall 1\leq i,j\leq n. \qquad (55)$$

Then, we introduce the symmetry of Riemannian metric as follows:

$$g^{ij}(X_t^\theta) \coloneqq g^{ij}(X_t^\theta)(\partial_i^t,\partial_j^t) = [g_{ij}]^{-1}(X_t^\theta)(\partial_i^t,\partial_j^t) = [g_{ij}]^{-1}(X_t^\theta)(\partial_j^t,\partial_i^t) = g^{ji}(X_t^\theta). \quad (56)$$

In other means, $g^{ij} = g^{ji}$ almost surely. Similarly, we can show the symmetry of Christoffel symbol $\Gamma_{jk}^i = \Gamma_{kj}^i$. Overall, the expectation of the following evaluations is equal to 1 almost surely:

$$\mathbb{E}_{\mu_{t\wedge\tau}^\theta}\left[\mathbf{1}\left\{\sum_{i,j,k}|\Gamma_{jk}^i(X_{t\wedge\tau}^\theta)|^2 \leq \frac{n^2(n+1)c_1}{2}\right\}\right]$$

$$= \mathbb{E}_{\mu_{t\wedge\tau}^\theta}\left[\mathbf{1}\left\{\sum_{j,k}|g^{jk}(X_{t\wedge\tau}^\theta)|^2 \leq \frac{n(n+1)c_1}{2}\right\}\right] = 1. \quad (57)$$

By combining (57), (54), and (55) with assumptions **(H4)** and **(H5)**, the proof is completed.

## D   Geometric Calculation

In this section, we provide the detailed information about geometric objects defined on the 2-Sphere $\mathbb{S}^2$, 2-Torus $\mathbb{T}^2$. Given the Riemannian manifolds equipped with Riemannian metric $(\mathcal{M}, g)$, we define three geometric objects including Christoffel symbol, Riemannian metric tensor, and its determinant to simulate the RNSDE.

Let us denote $\partial_i$ is the (spatial) partial derivative with respect to the $i$-th component on local coordinate. Then, Christoffel symbol is defined as follows:

$$\Gamma_{jk}^i(X_t^\theta) = \frac{1}{2}g^{ke}(X_t^\theta)\left[\partial_j g_{ei}(X_t^\theta) + \partial_e g_{ej}(X_t^\theta) - \partial_i g_{ij}(X_t^\theta)\right]. \qquad (58)$$

2-**Sphere.** The Riemannian metric and Christoffel symbol on 2-sphere are calculated as follows:

$$\text{Sphere } \mathbb{S}^2: \quad g_{ij}(X_t^\theta) = \begin{pmatrix} 1 & 0 \\ 0 & \sin^2 \vartheta \end{pmatrix}, \quad g^{ij}(X_{t \wedge \tau}^\theta) = \begin{pmatrix} 1 & 0 \\ 0 & \frac{1}{\sin^2 \vartheta} \wedge c_1 \end{pmatrix},$$

$$\Gamma_{jk}^i(X_{t \wedge \tau}^\theta) = \left[ \underbrace{\begin{pmatrix} 0 & 0 \\ 0 & -\sin \vartheta \cos \vartheta \end{pmatrix}}_{i=1}, \underbrace{\begin{pmatrix} 0 & \cot \vartheta \wedge c_1 \\ \cot \vartheta \wedge c_1 & 0 \end{pmatrix}}_{i=2} \right]. \tag{59}$$

2-**Torus.** Similar calculations follow on 2-torus.

$$\text{Torus } \mathbb{T}^2: \quad g_{ij} = \begin{pmatrix} R + r \cos \varphi & 0 \\ 0 & r^2 \end{pmatrix},$$

$$\Gamma_{jk}^i = \left[ \underbrace{\begin{pmatrix} 0 & -\frac{r \sin \varphi}{R + r \cos \varphi} \\ -\frac{r \sin \varphi}{R + r \cos \varphi} & 0 \end{pmatrix}}_{i=1}, \underbrace{\begin{pmatrix} \frac{(R + r \cos \varphi) \sin \varphi}{r} & 0 \\ 0 & 0 \end{pmatrix}}_{i=2} \right], \tag{60}$$

where we set $[R, r] = [1.0, 0.4]$ in our experiment on helix coils.

# E   Implementation Details

## E.1   Experimental Settings

We used a single GTX 2080 Ti GPU for all experiments. To train the RNSDE, we utilized the Adam optimizer with a learning rate of $10^{-3}$. We followed the identical network architecture suggested by prior work (9) in Appendix.6 of (9). The experimental results were obtained using open-source codes of the author's repositories:

- MCNF; `https://github.com/CUAI/Neural-Manifold-Ordinary-Differential-Equations`

- EMSRE; `https://github.com/katalinic/sdflows`

- RCPM; `https://github.com/facebookresearch/rcpm`

## E.2   Details about experiment on Vessel Route Dataset

**Mean Geodesic Errors.** We proposed the additional evaluation metric called MGE in Section 6 of the main paper. We specified the detailed formulation of MGE and related it with Wasserstein distance. Let $\mu, \nu$ be elements of Wasserstein space $\mathcal{P}_2(\mathcal{M}) := \left\{ \mu \, ; \, \mathbb{E}_{x \sim \mu} \left[ \frac{1}{2} \gamma^2(x, x_0) \right] < \infty, \forall x_0 \in \mathcal{M} \right\}$. Then we define the mean geodesic errors as follows

$$\textbf{MGEs}: \quad \frac{1}{2} \mathbb{E}_{(X_t^\theta \sim \mu_t^\theta, y \sim \nu_t)} \left[ \gamma^2(X_t^\theta, y_t) \right]. \tag{61}$$

By replacing $c(x, y) = \frac{1}{2} \gamma^2(x, y)$, mean geodesic errors induce the upper bound of Wasserstein distance:

$$\mathcal{W}_2^2(\mu_t^\theta, \nu_t) := \min_\Pi \mathbb{E}_\Pi \left[ c(x, y) \right] \leq \mathbb{E}_{(\mu_t, \nu_t)}[c(x, y)] := \frac{1}{2} \mathbb{E}_{(X_t^\theta \sim \mu_t^\theta, y \sim \nu_t)} \left[ \gamma^2(X_t^\theta, y_t) \right]. \tag{62}$$

While the task is to reconstruct the time-series on the ocean, we followed a similar protocol suggested in (9; 10) to evaluate the quality of generated sequences from models. Specifically, Compared to their works, we replaced the Euclidean geodesic error (*i.e.*, mean squared error, MSE) with the Riemannian geodesic error (*i.e.*, MGE) to reflect the accurate performance regarding the underlying geometry.

**Objective function.** In this experiment, the proposed method was trained to minimize the following objective function:

$$\min_\theta \int_0^T \mathcal{J}([A^*, B^*, \epsilon, t, \theta]) + \frac{1}{2} \mathbb{E}_{(X_t^\theta \sim \mu_t^\theta, y \sim \nu_t)} \left[ \gamma^2(X_t^\theta, y_t) \right] dt. \tag{63}$$

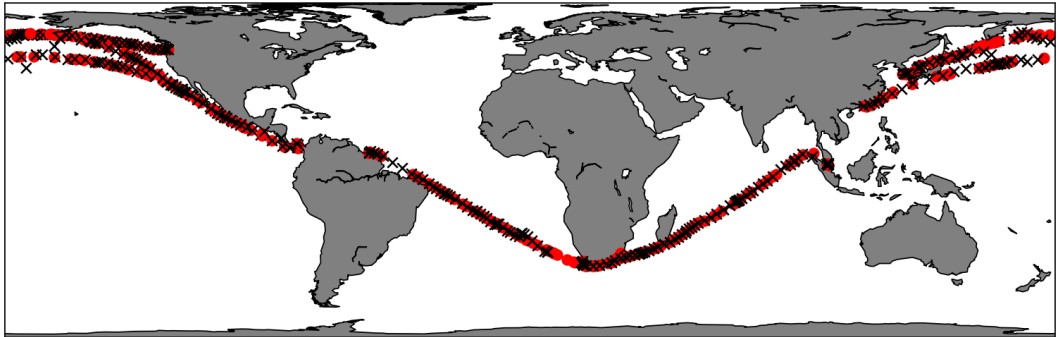

**Figure 1:** Reconstruction results on the Vessel Route dataset.

For fair comparisons with baseline methods (*i.e.*, Latent ODE, ODE-RNN), we modified the our original objective function to the form in (63).

**Qualitative Results on Vessel Route Dataset.** The vessel route dataset[1] contains sampled routes of 100 individual vessels on the ocean with 64 timestamps. For this experiment, we randomly sampled 5 stochastic trajectories among 100 different vessel routes starting from different continents. Fig.1 shows samples from reconstructed trajectories (**black**) and target trajectories (**red**). As shown in the figure, our model correctly follows the real vessel routes.

## F   Potential Theoretical Concerns

In this section, we answer potential questions about theoretical concerns of the proposed method, which are not dealt with in the main paper due to the page limit.

**1) Why do we focus on the embedded compact manifolds?** Despite the proposed method (*i.e.*, RNSDE) provides an universal framework for arbitrary manifolds, there is a theoretical concern for implementing the proposed RNSDE on non-compact manifolds owing to the "blowing-up" property of SDE (5). The solution to the RNSDE needs to exist for every given initial distributions with compact support. However, it is intractable in our framework to consider whether this condition is assured for every functions $\beta(\theta)$ and $W(\theta)$ regarding the neural network $\theta$. The future work will discuss about additional theoretical assumptions on this problem. Because our model manifolds (*e.g.*, sphere) assure the compactness, the proposed RNSDEs are well-defined for all $0 \le t < \infty$.

**2) Why do we consider the parameterization by considering the fixed chart?** It is well-known that the 2-sphere requires at least two charts to fully cover its surface. Let $f : \mathcal{M} \to \mathbb{R}$ be an arbitrary bounded continuous function. Then, we show the following calculation:

$$
\begin{aligned}
\int_{\mathbb{S}^2 \setminus \mathbf{P}} f(X_t^\theta) d\mu_t^\theta &= \int_{\mathbb{S}^2 \setminus \mathbf{P}} f(X_t^\theta) p_\theta(t, X_t^\theta) \sqrt{|\det G(x)|} dx \\
&= \int_{\mathbb{S}^2} f(X_t^\theta) p_\theta(t, X_t^\theta) \sqrt{|\det G(x)|} dx = \int_{\mathbb{S}^2} f(X_t^\theta) d\mu_t^\theta,
\end{aligned}
\tag{64}
$$

where $\mu_t^\theta = p_\theta(t, x) d\mathbb{V} = p_\theta(t, x) \sqrt{|\det G(x)|} dx$ and $\mathbf{P}$ is an arbitrary point on 2-sphere. The second equality in (64) holds because $dx$ can be considered as a Lebesgue measure on $\mathbb{R}^n$, and the expectation is taken over $\mathbb{S}^2$ excluding measure-zero set $\{\mathbf{P}\}$. Because the 2-Torus is diffeomorphic to the product of circles $\mathbb{S}^1 \times \mathbb{S}^1$ requiring 4 covers, similar calculation follows in the case of tori. The equalities in (64) show that utilizing a single chart in our method is acceptable. Contrary to our considerations, careful attentions are required when considering arbitrary Riemannian manifolds.

**3) Why do we consider the simple Euler-Maruyama scheme to simulate the RNSDE?** In conventional SDE-based methods, they utilized the SDE solver (7) which was suggested to simulate the stochastic paths of neural SDEs. Unfortunately, instability issue was addressed in recent study (3). They reported that SDE solver may induce unstable training in the computation of gradients. The RNSDE has similar problems during the training process, especially owing to the geometric calculations, where we discuss about gradient explosion problems in the proof C.5. In contrast to

---

[1]`https://www.noaa.gov/`

conventional SDE solvers that intrinsically bring complex inner procedures, the proposed method utilizes a **simple** Euler-Maruyama Scheme for simulating the RNSDE. The results show that desirable performance can be achieved regardless of the types of manifolds.