# OpenReview forum: "Riemannian Neural SDE: Learning Stochastic Representations on Manifolds"
_NeurIPS.cc/2022/Conference — NeurIPS 2022 Accept_

### Official Review · Reviewer_hLQB · 2022-07-09

**Rating:** 6
**Confidence:** 2
**Soundness:** 3 good
**Presentation:** 4 excellent
**Contribution:** 3 good

**Summary:**

The submission proposes an approach for modeling stochastic differential equations on Riemannian manifolds using neural networks called Riemannian neural SDE (RNSDE). This approach allows for learning a probability distribution over a manifold and modeling stochastic processes happening on the manifold. RNSDE uses neural networks to model the terms in the  local representation of an SDE and allows for probability density estimation using the dual metrics formulation. Experiments show that RNSDE outperforms some of the related deep learning baselines in a set of illustrative toy tasks and in several real-world benchmarks.

**Questions:**

1. It would be worth to consider diffusion models in related works, especially [1]
2. In the experiments with the Vessel route dataset, you choose Latent ODE and ODE RNN as the baselines. As far as I know the main feature of these models is that they can deal with irregular time series, i.e. where measurements are not equidistant wrt time. Does RNSDE capable to take this irregularity into account? If not, this comparison looks like comparing apples with oranges, where it is hard to make any conclusion why one model outperforms the others. I would highly recommend authors to add a comment on this in the text of the paper.
3. In the introduction section there’s a strong claim that conventional approaches which do not take into account geometry of the task are losing in their expressivity. However, there’s a little of validation for that in the experiments section. I would recommend to add this kind of conventional approaches as baselines to all the experiments to give a clear evidence of the importance of taking into account geometry of the data.

**Limitations:**

Authors clearly state  that they focus only on Euclidian manifolds while there’re other domains (non-compact Lie groups, product spheres) important for downstream applications to consider. There’re no discussion of the negative social impact provided.

**Strengths And Weaknesses:**

### Originality

According to the provided literature review, the submission proposes a novel approach to modeling SDEs on manifolds. Authors clearly state the difference between their method and related works, highlighting that RNSDE requires less trainable components to model evolution of the density and claiming that it provides potentially more flexible solution than some other restricted methods. However, the literature review does not cover diffusion models which seem to be very relevant to the proposed method, especially in its application to density estimation. Specifically, there’s a relevant work [1] studying diffusion models on Riemannian manifolds which would be worth to compare with or at least to mention as a competitive work and highlight the key differences.

### Quality

The submission is technically sound and complete with a solid theoretical basis and good enough experiments. Authors’s claims are mostly supported by experimental results and most of the baseline methods are appropriate.

### Clarity

The presentation of the method is very decent, authors introduce all the important concepts and provide preliminaries required for understanding the method. They also support some of the involved concepts and derivations with intuitive explanations which are very helpful for a broader machine learning audience. Moreover, informative illustrations helps a lot to make sense of the method and experiments. However, a solid background in manifold theory is required to fully understand the method and its theoretical basis.

### Significance

The proposed method addresses an important task of modeling SDEs on manifolds in an original way surpassing some of the existing methods. The literature review and experimental comparison suggests that the proposed method is fairly significant.

[1] De Bortoli, Valentin, et al. "Riemannian score-based generative modeling." arXiv preprint arXiv:2202.02763
 (2022).

---

> ### Author Response · Authors · 2022-08-01
> **Response to Reviewer hLQB**
>
>
> We thank the reviewer for their kind feedback on our work. We addressed all the comments in the updated version of our manuscript and provided a point-by-point response below.
>
> $\textbf{[Comparison to the prior work]}$
>
> For a comprehensive discussion, we provided a detailed comparison in the [general response] section. Please refer to the general response section.
>
> $\textbf{[Concerns about irregularity]}$
>
> In the vessel route experiment, time stamps are essentially $\textbf{non-uniform}$ and $\textbf{irregular}$ as the dataset contains information of vessel navigation information (e.g., geographic coordinate) acquired from separate time scales. To make a reconstruction using this dataset, our method supports the irregular temporal setting. We followed the identical protocol suggested in the previous work [A] to train baselines where the irregular events were the standard assumption. Since their model (CSDE-TP, [A]) is defined on the ambient Euclidean space $(\mathbb{R}^3, d_E)$, we substitute the Euclidean distance to the Riemannian counterpart $(\mathbb{S}^2, d_{\mathbb{S}^2})$.
>
> $\textbf{[Empirical Validation for Representability]}$
>
> Mathematically, any points on the unit sphere have a unit norm. Thus, the following condition should be satisfied if the neural dynamics model respects the underlying geometry of the sphere.
> \begin{equation}
>     X = [x^1, x^2, x^3]^T, \quad \sqrt{ (x^{1})^2 + (x^{2})^2 + (x^{3})^2} = 1,
> \end{equation}
> where $X_t = [x_t^1, x_t^2, x_t^3]$ are the network outputs at time interval $t \in [0, T]$. This mathematical constraint is somewhat artificial while neural networks generally produce vectors in $\mathbb{R}^3$. In this regard, prior works applied the additional projection function to impose the global structural constraint to satisfy the constraint. Regarding the discussion to train baseline Euclidean methods, we utilized a function called a $\textbf{stereographic projection}$ that projects the vectorial network outputs onto the sphere. After the projection procedure, we trained the baseline methods (Latent ODE, ODE-RNN) by replacing the usual Euclidean distance with the Riemannian one to calculate a Gaussian-type log-likelihood suggested in [B]. The key point in this experiment is whether prior works perform well even after the geometric projection. Table 4 shows that our method outperforms prior works and Euclidean opponents may lose expressivity on the sphere.
>
> -------
>
> [A] Neural Markov Controlled SDE: Stochastic Optimization for Continuous-Time Data, ICLR 2022.
>
> [B] Latent Ordinary Differential Equations for Irregularly-Sampled Time Series, NeurIPS 2019.

---

### Official Review · Reviewer_mk19 · 2022-07-11

**Rating:** 6
**Confidence:** 4
**Soundness:** 4 excellent
**Presentation:** 4 excellent
**Contribution:** 3 good

**Summary:**

This paper introduces a neural network method to learn and represent stochastic differential equations on manifolds. Applications are presented for density modeling on manifolds and learning latent paths on manifolds.

**Questions:**

My questions are covered in the weaknesses section. In particular, I would like to see a comparison with Moser Flow and a discussion with the Riemannian Score Based Modeling Paper.

**Limitations:**

Yes

**Strengths And Weaknesses:**

Strengths
-----------
+ The paper is presented very well. I appreciated the clear presentation, precise mathematical language, thorough writing, and informative figures.
+ The paper is very technically solid, as the core difficulty of extending Neural SDEs to manifolds is aptly handled with the correct Riemannian-geometric constructs.
+ The experiments are pretty well done, and the Vessel Route dataset is an interesting contribution to this space.

Weaknesses
--------------
- There is work that slightly predates this one on extending SDEs to manifolds in the context of scored based generative models and diffusion (https://arxiv.org/abs/2202.02763). While I believe that the two works are different, the above work should be included in the discussion.
- For the density modeling experiments, some baselines are missing. In particular, I believe there should be a comparison with Moser Flow.

Verdict
--------
I generally lean on accepting the paper, as I think that it is a solid contribution to the space. However, I would want some more discussion with the baselines and previous methods mentioned above (as I think they sufficiently predate the work).

---

> ### Author Response · Authors · 2022-08-01
> **Response to Reviewer mk19**
>
> Thanks a lot for your valuable feedback. Also, we thank you for specifying the strength of our paper. We revised our manuscript accordingly and provided a detailed response to the raised comments.
>
> $\textbf{[Comparison to the prior work]}$
>
> For overall discussion, we clarified an elaborate comparison with prior work in the [general response] section. We specified the difference in motivation, technical solutions, etc in the section and our main paper. Please kindly refer to the general response section.
>
> $\textbf{[Additional Experiment]}$
>
> As the reviewer requested, we set an additional experiment including the result of Moser Flow on the density estimation task. For the experiment, we thoroughly utilized the original experimental setting of Moser Flow. To train Moser Flow, 6 hidden layers of MLP with 512 neurons was parameterized, and training epochs and learning rate were set to be identical to the original settings. We referred to experimental settings from their public code implementation [https://github.com/noamroze/moser_flow].
>
> |Methods    | 8-shapes  | Two moons  | Spiral  |
> |--:|---|---|---|
> | Morser Flow  | 5.81  | 6.10  | 7.56  |
> | RNSDE | $\mathbf{5.67}$  | $\mathbf{5.66}$  | $\mathbf{6.97}$  |
>
> We evaluated the density estimation performance with the $2$-Wasserstein distance $\mathcal{W}_2$ ($\times 10^{-2}$) on each synthetic dataset. The above table shows that the proposed method still outperforms the Moser flow. As shown in Table above, Moser Flow performed well, especially on \textit{8-shapes}, but still our model surpassed the method on other densities. In the revised manuscript, we will add the results of Moser Flow.

---

### Official Review · Reviewer_rDVp · 2022-07-17

**Rating:** 6
**Confidence:** 3
**Soundness:** 3 good
**Presentation:** 3 good
**Contribution:** 2 fair

**Summary:**

This paper proposes a method to express the stochastic representation over manifolds, using a novelly defined Riemannian Neural stochastic differential equation (RNSDE), the method is theoretically well-analyzed at each step, empirically RNSDE outperforms several baselines on some simple tasks.

**Questions:**

Q1: I understand that Euclidean Neural SDE gives results not on the manifold, but a very naive approach is to enforce a hard constraint (e.g. projection to the manifold) after each Euclidean neural SDE step, or something as a first order retraction? One can also move along local frameworks to get a trajectory on the manifold, do you include this baseline to compare with? Both performance and computations

Q2: I’m curious what’s the choice of the stopping time tau could affect in practice? How large it needs to be, how severe the gradient explosion is and how to choose this value?

**Limitations:**

Mostly discussed above, the practical usage seems to be rare, and the computations are heavy, limited to a small set of manifolds and low dimensions.

**Strengths And Weaknesses:**

I’m not an expert on neural SDE, so some points maybe stupid :)
S1: the method is theoretically sound in most steps, except the learning step and some approximations;
S2: the extension from Euclidean neural SDE to Riemannian Neural SDE makes it easer to understand and follow;
S3: Empirically the proposed RNSDE outperforms some previous baselines on tasks including generative modeling, interpolation and reconstruction.

W1: I think this paper is not that ‘readable’ for general NeurIPS audience, but only experts in this area, currently it’s heavy math, I would suggest add more explanations/backgrounds and intuitions to the text, for example the Eells-Elworthy-Malliavin interpretation of the diffusion process, and what each step in the Scheme is doing, this could significant help abstract more readers into this area.

W2: RNSDE incurs more computations than the Euclidean case, including metric tensors, christoffee symbol and etc, which has to be computed for every local neighborhood as moving along the path, I would like to see some computation burden analysis or empirical results of RNSDE, plus compared to some baselines, particularly to one I mention in next question part.

W3: The practical usages/applications of this method looks rare to me, what’s more,  the manifolds are also simple, though S^n is claimed to be one model, I only see n=2 case, and with a torus model, is it possible to test on high dimensional manifolds?

---

> ### Author Response · Authors · 2022-08-01
> **Response to Reviewer rDVp**
>
> Thank you for pointing out the strengths of our paper. Moreover, we thank the reviewer for the detailed review and insightful comments. Below, we addressed the comments raised by the reviewer.
>
> $\textbf{[Computational Complexity, Scalability]}$
>
> Due to the computational complexity possibly raised by geometric objects, we restricted our interests to the case where two geometric objects are analytically tractable. Especially, we require two geometric objects including $\textbf{Riemannian metric tensor}$ $\{ g_{ij} \}$, and the $\textbf{generalized distance function}$ $c(x,y)$ (called a cost function in optimal transport literature). The former is used to define the proposed dynamical system on manifolds, and the latter is for the dual optimal transport problem. Thankfully, the additional computational costs are insignificant during the training and inference time as we priorly compute these objects using the mathematical definition introduced in Appendix (59) and (60). To address the scalability issue, we set an additional experiment by increasing the dimensionality for the product copies of manifolds, $\prod_{k}^K (\mathbb{S}^2)^{k=1} = \mathbb{S}^2 \times \cdots \times \mathbb{S}^2$. The following table shows the average training time per epoch given the different dimensionality $K \in \{1, 4, 8, 16\}$.
>
> |K=1|K=4|K=8|K=16|
> |--:|---|---|---|
> |0.82|1.16|1.83|2.07|
>
> As shown in the table, the additional computational burden is less than linear growth. We checked that the number of temporal states for discrete Euler-Maruyama Scheme is the primal factor that causes high computational costs as similar to conventional neural dynamical models.
>
> $\textbf{[Geometric Constraint by Projection]}$
>
> If one projects the representation of Euclidean neural SDE in (1) onto the sphere by using length scaling $X'_t = P(X_t) \coloneqq X_t / || X_t ||_E$, the diffusion part in the second term is suppressed. Eventually, the projected representation $X'$ may lose the diffusive property if the method continues the projection during the stochastic propagation (Euler-Maruyama scheme). This argument can be easily shown by the Ito's formula:
> \begin{equation}
>     dX'_t = [ \textbf{ drift } ] dt + \beta D^2 P(X_t) dW_t \approx [ \textbf{ drift } ] dt + \beta D \left( \frac{1}{\||X_t||_E}\left( I_d - \frac{\langle X_t, X_t\rangle}{ ||X_t||_E^2}\right) \right)dW_t,
> \end{equation}
> where $D^2$ and $D$ are Euclidean differentials of orders $2$ and $1$, respectively.
> Note that we regard $X_t$ as a solution to Euclidean SDE with respect to Ito's integral for simplicity. Since the derivative in second term is approximately $\approx O(||X||^{-2})$, the diffusive behavior is suppressed after projection.
>
>
> There exist only few known global maps such as length scaling that can project vectors in the ambient Euclidean space $\mathbb{R}^3$ to a specific type of model manifolds such as a sphere $\mathbb{S}^2$. For the generality, the key point in applying manifold theory to neural SDE is to define Riemannian counterpart in the local sense.
>
> In the literature [C], they already proposed a similar but concrete concept that applies the projection method to respect the geometric constraint $||X||_E = 1$. In particular, they defined the diffusion on the sphere by using a global projection that transforms Euclidean vectors onto the spherical tangent space (not the model sphere) and enjoys the diffusion property:
>
> $dX_t^i = \left( \delta_{ij} - X_s^i X_s^j \right) \circ dW_s^j, \quad X_0 \in \mathbb{S}^2$
>
> Note that the above spherical Brownian motion corresponds to a specific case of the second term in (2) to construct the diffusion on the sphere. Contrary to the above example, we adopt the general point of view to define the diffusion process as horizontal representations.
>
> $\textbf{[Effect of Stopping time in practice]}$
>
> We empirically selected a hyper-parameter $c_1$ in (51) by checking the probability of occurrence of infeasible numbers in experiments. In general, $c_1 = 10^3$ was enough to stabilize the training scheme. We observed that the co-metric tensor occasionally produced infeasible values ($\approx 10^8 \gg c_1)$ with a low probability in the spherical experiments. If these unexpected values are fed into the proposed RNSDE, then it quickly falls into failure mode during the Euler-Maruyama scheme. As aforementioned in Appendix L186~196, this phenomenon is predictable because of the specific form of suggested metric tensors.
>
> --------
>
> [C] An introduction to the analysis of paths on a Riemannian manifold, Daniel W. Strook

---

### Author Response · Authors · 2022-08-01
**General Response to all Reviewers**

We are glad to inform that every reviewers appreciated the contributions and acknowledged the strength of our paper. In this section, we compare our model with prior work [D] and give a brief explanation and comparison. The contents in this comment will be added in an additional content page for the camera-ready version.

$\textbf{[Comparison to the prior work]}$

In [D], they suggested the time-reversed diffusion process and the corresponding score-matching framework for generative modeling. In contrast to our approach, they proposed a geodesic random walk to propagate the stochastic dynamics which naturally ensures the intrinsic geometric structure. Then, the denoising score-matching was applied by eigen-decomposition of the transition probability as a heat kernel with respect to the Fokker-Planck equation in Appendix (10). In generative modeling, our method follows a different methodology while our interest lies in matching the endpoint of the probabilistic state by setting a static bridge via log-Sinkhorn.

------

[D] Riemannian Score-Based Generative Modeling, 2022, preprint.

---

### Meta-Review · Area_Chair_cEms · 2022-08-27

**Recommendation:** Accept
**Confidence:** Less certain

**Metareview:**

There was a consensus towards weak acceptance among all the reviewers, and I agree with this consensus. This paper solves an important problem of applying SDEs to manifolds. It is clearly written, and all the reviewers agree that the claims are well-supported by strong experimental results. On the other hand, this clarity of writing perhaps relies overmuch on familiarity with the area, and some effort should be made to smooth out the presentation for a general NeurIPS audience. Beyond this, the weaknesses pointed out by the reviewers were well addressed by the author response, including an additional experimental result that provides a comparison to Moser Flow: there is no strong unaddressed weakness that would merit rejection. I think that the manifold learning community, as well as the Neural ODE community more broadly, at NeurIPS will find this work interesting and useful, as it expands the range of methods they can apply on manifolds. As such, I lean towards accepting this paper.

**Award:**

No

---

### Decision · Program_Chairs · 2022-09-14

Accept